# A path towards high lithium-metal electrode coulombic efficiency based on electrolyte interaction motif descriptor

Ruhong Li [1,2,9], Xiaoteng Huang[1,9], Haikuo Zhang[1], Jinze Wang[1], Yingzhu Fan[3], Yiqiang Huang[1], Jia Liu [4], Ming Yang[5], Yuan Yu[6], Xuezhang Xiao [1], Yuanzhong Tan[7], Hao Bin Wu [1], Liwu Fan [4], Tao Deng [8], Lixin Chen [1], Yanbin Shen [3] & Xiulin Fan [1] ✉

The fundamental interactions and the as-derived microstructures among electrolyte components play a pivotal role in determining the bulk and interfacial properties of the electrolytes. However, the complex structure-property relationships remain elusive, leading to uncontrollable physicochemical characteristics of electrolytes and unsatisfied battery performance. Herein, we propose two interaction motif descriptors to quantify ion-solvent interactions spanning electrostatic to dispersion regimes. These descriptors are highly relevant to salt dissolution, phase miscibility, and electrode-electrolyte interface chemistries. Guided by the principle of minimizing ion-solvent and solvent-solvent interactions while ensuring sufficient salt dissociation, a representative electrolyte, *i.e.*, lithium bis(fluorosulfonyl)imide dissolved in trimethyl methoxysilane and 1,3,5-trifluorobenzene with a molar ratio of 1:2.5:3.0, is designed, which achieves ~99.7% (±0.2%) Li plating/stripping Coulombic efficiency and endows 4.5 V Li∥LiCoO$_2$ with 90% capacity retention after 600 cycles at 0.2 C/ 0.5 C charge/discharge rate. Notably, Cu∥LiNi$_{0.5}$Co$_{0.2}$Mn$_{0.3}$O$_2$ pouch cells with this electrolyte sustain over 100 stable cycles. By establishing quantitative relationships between interaction motifs and electrolyte functionalities, this work provides a universal framework for rational electrolyte design, paving the way for highly reversible lithium metal batteries.

Lithium metal batteries (LMBs) hold great promise for achieving twice the energy density of current lithium-ion batteries due to the high capacity (3860 mAh g$^{-1}$) and lowest potential of Li metal among the possible negative electrode materials[1–3]. However, the practical implementation of Li metal faces challenges such as Li dendrite growth and low Coulombic efficiency (CE), primarily attributed to the fragile solid electrolyte interphase (SEI) undergoing continuous destruction and reconstruction[4–6]. The interfacial chemistry, governing the

[1]State Key Laboratory of Silicon and Advanced Semiconductor Materials, School of Materials Science and Engineering, Zhejiang University, Hangzhou 310027, China. [2]ZJU-Hangzhou Global Scientific and Technological Innovation Center, Zhejiang University, Hangzhou 311215, China. [3]i-Lab, CAS Center for Excellence in Nanoscience, Suzhou Institute of Nano-Tech and Nano-Bionics (SINANO), Chinese Academy of Sciences, Suzhou 215123, P. R. China. [4]State Key Laboratory of Clean Energy Utilization, School of Energy Engineering, Zhejiang University, Hangzhou 310027, China. [5]Science and Technology on Power Sources Laboratory, Tianjin Institute of Power Sources, Tianjin 300384, China. [6]Zhejiang Provincial Key Laboratory of Fiber Materials and Manufacturing Technology, Zhejiang Sci-Tech University, Hangzhou 310018, China. [7]Zhejiang Xinan Chemical Industrial Group Co. Ltd, Hangzhou 311600, P. R. China. [8]China-UK Low Carbon College, Shanghai Jiao Tong University, Shanghai 201306, China. [9]These authors contributed equally: Ruhong Li, Xiaoteng Huang. ✉e-mail: xlfan@zju.edu.cn

formation of passivation layers is closely related to the solvation structures and underlying intermolecular non-covalent interaction (NCI)[7,8]. These complex and intertwined interactions between electrolyte components give rise to some phenomena such as changes in solution structure and phase transitions, presenting a formidable challenge in the rational design of electrolytes from a bottom-up approach[9–11].

An electrolyte system typically involves intricate interactions between Li⁺, solvent molecules, and anions[12,13]. These interactions determine the Li⁺ solvation configuration, lithium salt solubility, and the electrolyte-electrode interface properties[14–16]. For instance, the interactions involving Li⁺ dictate the dissociation of lithium salt, thereby regulating the solvation structure and the formation of passivation films[17–19]. This structure-function relationship enables the control of desirable interfacial reactions by tuning the interplay of electrolyte components[20]. Hence, significant efforts have been devoted to comprehensively understanding the interactions and their impact on electrolyte properties in recent years[21–23]. These efforts encompass investigations into the anion-cation interactions for ion pairs and aggregates formation[24,25], anion-solvent interactions for solvents and radical solvation[26–28], and cation-solvent interactions for solvents reductive stability and cation (de)solvation behaviors[29–31]. Among them, the highly concentrated electrolytes (HCEs)[32,33] and localized highly concentrated electrolytes (LHCEs)[34,35] have emerged as promising systems, which exhibit high Li metal plating/stripping efficiencies and improved oxidation stabilities due to the abundance of contact ion pairs (CIPs) and aggregates (AGGs)[36–38]. State-of-the-art LHCEs have achieved a CE exceeding 99% through the regulation of anion-cation-solvent pairs interactions[39,40]. Nevertheless, numerous existing strategies utilize imprecise terminology to characterize these interactions, which inadequately distinguishes between ion-solvent and solvent-solvent binding mechanisms. This ambiguous classification induces misinterpretations of critical phenomena including lithium salt dissolution limitations, liquid-phase segregation behavior, and ultimately obstructs the systematic development of advanced electrolytes[41–43]. Therefore, the establishment of comprehensive concepts to discern complex interactions is highly anticipated as it would accelerate progress in current electrolyte engineering and realize next-generation high-energy LMBs.

In this work, we implement two structurally distinct descriptors for Li⁺ and FSI⁻ species to address their fundamentally divergent solvation behaviors and chemical interactions within the electrolyte system. Li⁺, as a hard Lewis acid, primarily interacts with solvent molecules through strong electrostatic interactions and directional coordination bonds, necessitating a descriptor that captures its solvation environment and binding strength. The ratio of electrostatics to induction $E_{ele}/E_{ind}$ between solvent and Li⁺ reflects the relative dominance of electrostatic versus inductive forces in shaping the cation solvation structure. In contrast, anions exhibit more complex solvation dynamics governed by synergistic effects including polarizability, charge delocalization, and ion-pairing tendencies, requiring a distinct descriptor to accurately capture these effects. The ratio of electrostatics to dispersion $E_{ele}/E_{dis}$ between solvent and FSI⁻ evaluates the interplay between long-range electrostatic forces and short-range dispersion interactions in anion-solvent interactions, which directly correlates with the anion's ion-cluster dynamics, thereby modulating its reduction susceptibility and subsequent participation in SEI formation processes. The dispersion-dominated FSI⁻-solvent interactions stabilize the solvation shell, reduce the dissociation of Li⁺ solvation structures, and extend the lifetime of ion pairs, further enhancing the stability and performance of the electrolyte. As a proof of concept, 1,3,5-trifluorobenzene (FB₁₃₅) and trimethyl methoxysilane (TMOS) were cherry-picked to formulate a LiFSI-2.5TMOS-3.0FB₁₃₅ electrolyte (where the numbers indicate molar ratios). The designed electrolyte can achieve 99.7% (±0.2%) Li plating/stripping CE and cycle stably for

over three months at a high areal capacity of 3 mAh cm⁻² in a full platting/stripping test. Moreover, the 20-μm-Li‖2.3-mAh-cm⁻²-4.5-V-LiCoO₂ (LCO) cells with the designed electrolyte achieve capacity retention (CR) of 90% over 600 cycles, outperforming the commercial carbonate electrolyte and other Li metal-friendly electrolytes. Notably, the cells with the LiFSI-2.5TMOS-3.0FB₁₃₅ electrolyte deliver a stable cycling of >100 cycles under practical conditions.

## Results and discussion
### Electrolyte interaction motif and design principle

The rational design of prospective electrolyte systems necessitates the primary selection of electrochemically stable solvents with cathodic compatibility, as quantitatively assessed through their reduction potentials ($E_{reduction}$ *vs*. Li⁺/Li, Supplementary Fig. 1 and Supplementary Table 1). Furthermore, the solvation structure is thought to directly determine the chemical components of the SEI, thus, impacting Li deposition morphology and CE. In a typical electrolyte system, the interaction between ions and solvents serves as the foundation for the formation of solvation structure. Within this framework, the local ordered coordination structure is primarily regulated by the strong cation-solvent interactions, while isolated solvation species exhibit disordered distribution at long range due to the weak interactions among coordinated and free solvent molecules, as well as anions (Fig. 1a)[44,45]. To evaluate the strength of the potential ion-solvent interactions, interaction energies and binding free energies (Supplementary Fig. 2) were meticulously quantified at a high computational accuracy level. The findings, as summarized in Supplementary Table 2, demonstrate a significant disparity in the interaction energies between charged Li⁺-solvent complexes (−10-−100 kcal mol⁻¹) and FSI⁻-solvent ones (0-−10 kcal mol⁻¹). This observation agrees well with the extensively discussed cation solvation theory, corroborating the substantial strength discrepancy between Li⁺ and FSI⁻ interactions with solvents. In contrast to the distinct strength relationship in Li⁺-solvent interactions, the interactions involving the majority of FSI⁻-solvent complexes demonstrate relatively proximal values. This characteristic, to a certain extent, poses a challenge in precisely quantifying differences among these interactions.

To shed light on the electronic characteristics governing non-covalent interactions within electrolyte ingredients and identify the primary stabilizing factors, we employed symmetry-adapted perturbation theory (SAPT) to compute interaction energies with mean absolute error <0.85 kcal mol⁻¹ (Supplementary Note 1 and Supplementary Fig. 3)[46,47]. In the framework of the SAPT approach, the ion-solvent interaction energy can be divided into various physically meaningful components, i.e., electrostatic, exchange-repulsion, induction, and dispersion. As shown in Supplementary Tables 3, 4, the cumulative contributions from attractive electrostatics, induction, and dispersion terms outweigh the repulsive exchange-repulsion contribution, leading to negative interaction energies. Moreover, the interactions related to Li⁺ ions are predominantly governed by electrostatic and induction, whereas the interactions associated with FSI⁻ are mainly characterized by electrostatics and dispersion (Supplementary Fig. 4). To facilitate a more precise classification of ion-solvent interaction motifs, we employ descriptors that define the distribution of interactions between molecules and Li⁺ (Ratio of electrostatics to induction, $E_{ele}/E_{ind}$ (Li⁺)), and between molecules and FSI⁻ anions (Ratio of electrostatics to dispersion, $E_{ele}/E_{dis}$ (FSI⁻)). As illustrated in Fig. 1b, regions II and IV are populated by molecules with $E_{ele}/E_{ind}$ (Li⁺)>1.3, indicating a dominance of ion-dipole interactions over ion-induced dipole interactions. Molecules in these regions generally exhibit stronger interactions with Li⁺, a characteristic feature of solvents in dual-solvent systems. In contrast, molecules in zones I and III, with lower $E_{ele}/E_{ind}$ (Li⁺), are more likely to act as diluents due to weaker overall interactions (Fig. 1c). Notably, an $E_{ele}/E_{dis}$ (FSI⁻) value less than 1.0 indicates interaction with FSI⁻ dominated by dispersion rather than

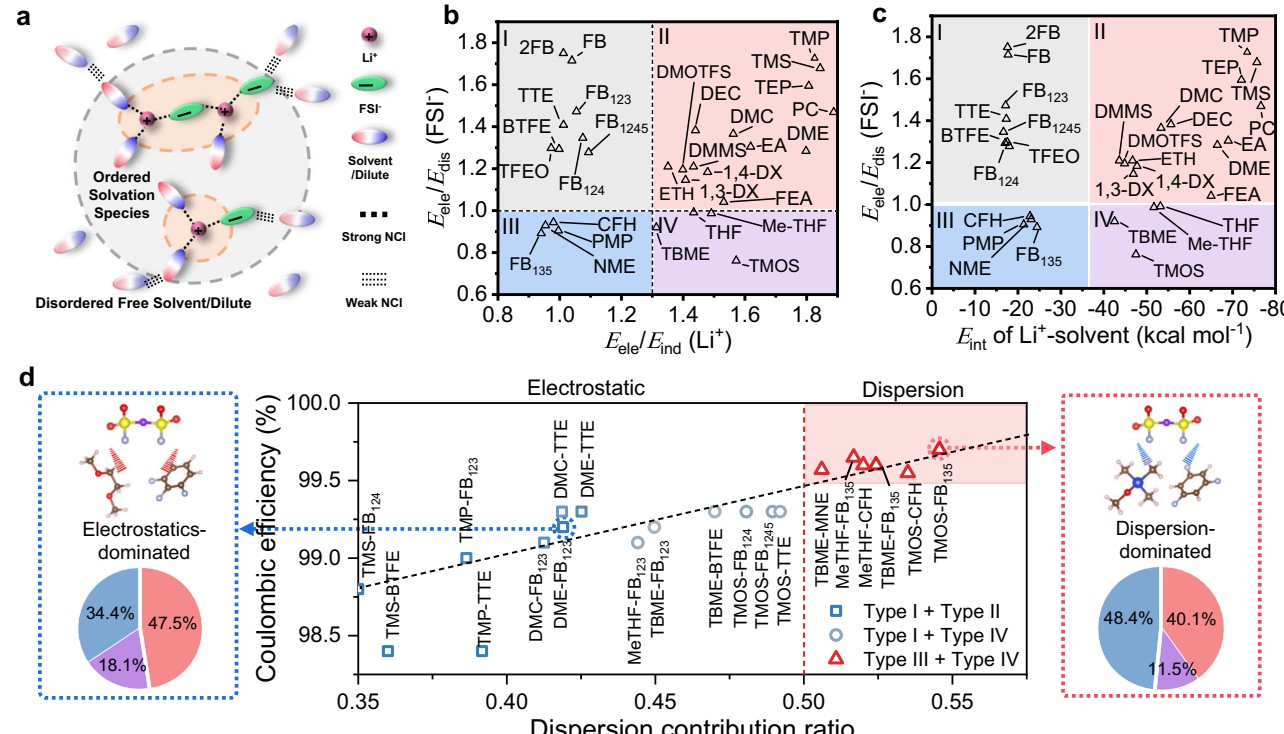

**Fig. 1 | Electrolyte design strategies. a** Schematic illustration of the ion-solvent interaction regulated solvation structure. **b** The molecular design diagram of $E_{ele}/E_{dis}$ (FSI⁻) versus $E_{int}$ (Li⁺). **c**, The molecular design diagram of $E_{ele}/E_{dis}$ (FSI⁻) versus $E_{ele}/E_{ind}$ (Li⁺). Molecular structures and corresponding abbreviations of the investigated solvents are shown in Supplementary Fig. 1. **d** Relationship between the dispersion contribution ratio and Coulombic efficiency of electrolytes composed of different types of solvents, schematic interactions of the diluent-FSI⁻-solvent pairs (left is FB₁₂₃-FSI⁻-DME, right is FB₁₃₅-FSI⁻-TMOS, respectively) and corresponding ratios of three non-covalent interaction (NCI) components, which was obtained by processing data in Supplementary Fig. 4. The red, purple and blue colors in the pie chart represent electrostatic force, inductive force and dispersion force respectively. Color code: red: O atom, yellow: S atom, purple: N atom, cyan: F atom, blue: Si atom, tan: C atom, white: H atom.

electrostatics, which divides the diluent area into zones I and III, and the solvent area into zones II and IV. Specifically, the interactions between molecules located at zones III/IV and FSI⁻ are governed by dispersion forces, while the molecules of zones I/II are primarily coordinated with FSI⁻ by strong electrostatic forces. By examining various representative electrolytes, we elucidated the interaction characteristics and corresponding Li plating/stripping CE, as displayed in Fig. 1d and Supplementary Table 5. In these solvent-diluent pairs, the primary interactions gradually transition from electrostatics to dispersion. This ratio can reflect the solvation energy to a certain extent[48], but for weakly solvated electrolyte systems, this ratio can be more accurately correlated with CE (Supplementary Fig. 5). All the investigated dispersion-dominated electrolytes showcase remarkable CE of ≥99.5%, notably highlighted by the performance of TMOS-FB₁₃₅ pair, achieving an CE of approximately ~99.7%. Specifically, in the TMOS-FB₁₃₅ pair (Fig. 1d), the proportion of dispersion (48.4%) has exceeded that of electrostatics (40.1%). Therefore, the calculated interaction ratio proves to be highly effective, serving as a robust metric for classifying a wide range of solvent molecules. This approach offers valuable insights that can guide the practical design and characterization of electrolytes. The efficacy of interaction ratios complemented by energetics analyses allows for a more accurate prediction of ion-cluster dynamics and interfacial regulation mechanisms, offering a promising perspective for electrolyte design.

## Solvation structure and interfacial reaction simulation

By combining two representative solvents (1,2-dimethoxyethane (DME) in zone II and TMOS in zone IV) and diluents (1,2,3-trifluorobenzene (FB₁₂₃) in zone I and FB₁₃₅ in zone III), we elucidated the solvation structures of four electrolytes using both classical and ab

initio molecular dynamics (MD) simulations. Subsequent benchmarking of the electrolyte structure and key physical parameters confirmed that the atomic-charge-scaled approach produced reliable results, showcasing comparable values to those derived from polarizable and quantum-based force fields (Supplementary Fig. 6 and Supplementary Tables 6, 7). The solvation structure and dynamics of various electrolytes were further analyzed. The average statistical size of the solvated clusters is 14.3 Å (where the numbers represent the anions in the solvated clusters) for LiFSI-2.5TMOS-3.0FB₁₃₅, which is larger than other electrolytes (Fig. 2a and Supplementary Fig. 7). The autocorrelation function results suggest that the lifetime of Li⁺-FSI⁻ pairs in the FB₁₃₅ system is longer than that in the FB₁₂₃ system (Fig. 2b and Supplementary Fig. 8). This relationship can be attributed to the role of dispersion-dominated solvents in modulating ion-pairing and aggregation dynamics. The high proportion and long lifetime of AGGs in LiFSI-2.5TMOS-3.0FB₁₃₅ electrolyte is consistent with Raman peak deconvolution analyses (Fig. 2c and Supplementary Fig. 9). The heat map analysis reveals the two most probable solvation structures in the LiFSI-2.5TMOS-3.0FB₁₃₅ system: 1TMOS-Li⁺-2FSI⁻ and 1TMOS-Li⁺-3FSI⁻, where most Li⁺ is surrounded by a TMOS molecule and more than one FSI⁻ anions (Fig. 2d and Supplementary Fig. 10). This can also be verified by the smaller donor number value of TMOS, while FB₁₃₅ with a relatively high dielectric constant is more like a medium that can adjust the solvation structure dynamics[49]. The formation of stabilized anion aggregation clusters (AGGs) with extended lifetimes (13.6 ps) maintains elevated local FSI⁻ concentrations, preferentially directing anion-derived SEI formation through selective decomposition pathways. This mechanism effectively suppresses solvent reduction while establishing inorganic-dominated interphases, thereby fundamentally enhancing LMA compatibility. Moreover, the phase separation phenomenon is

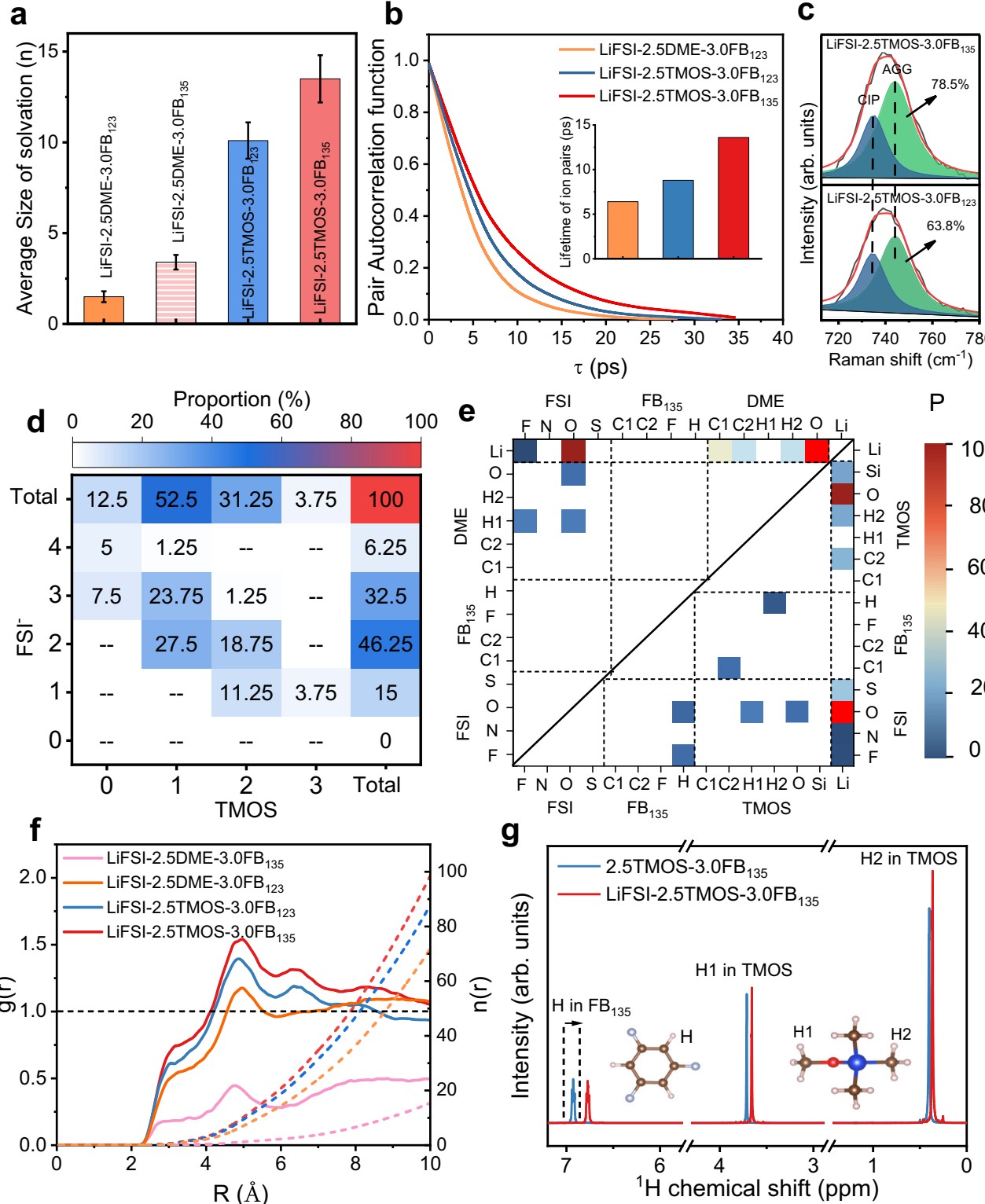

**Fig. 2 | Solvation structure and dynamics of the electrolytes. a** The average size of solvation clusters in electrolytes, which is defined as the average number of Li⁺ cations in the solvation cluster, and the error bar of size is defined as the difference between the maximum and minimum in the time-averaged values in MD simulation. **b** Li⁺-FSI⁻ pair existence autocorrelation function of different electrolytes with a cut-off distance of 2.5 Å. **c** Raman spectra and peak deconvolution results. **d** Heatmap showing the proportion of corresponding solvation structure. **e** The intermolecular

interaction matrix for LiFSI-2.5DME-3.0FB₁₃₅ (upper left) and LiFSI-2.5TMOS-3.0FB₁₃₅ (bottom right). The normalized probability density (P) of intermolecular connections varies from ~ 0 to 100 as coloring from purple to red. **f** radial distribution function for N (FSI⁻) versus H (FB₁₃₅) or H (FB₁₂₃) in various electrolytes. **g** ¹H NMR spectra of LiFSI-2.5TMOS-3.0FB₁₃₅ electrolyte and 2.5TMOS-3.0FB₁₃₅ solvent, insets show the ball-and-stick model of molecules FB₁₃₅ and TMOS. Color code: red: O atom, cyan: F atom, blue: Si atom, tan: C atom, white: H atom.

observed in LiFSI-2.5DME-3.0FB$_{135}$ electrolyte (Supplementary Fig. 11). The MD simulation snapshots also confirm the poor miscibility of FB$_{135}$ with DME and a predominance of LiFSI in the DME-rich region. The intermolecular interaction matrix analysis indicates that FB$_{135}$-related interactions are negligible in LiFSI-2.5DME-3.0FB$_{135}$ electrolyte (highlighted by the red border in Fig. 2e). The radial distribution function (RDF) results demonstrate that the solvation sheath of Li$^+$ predominantly comprises coordinating FSI$^-$ anions and DME/TMOS solvents (Supplementary Fig. 12). Notably, the pronounced interaction between FB$_{135}$ and anions can be observed in Fig. 2f. The complex yet subtle intermolecular forces in different electrolytes were further probed using nuclear magnetic resonance (NMR). The chemical shift

of H in FB$_{135}$ shifts up-field in the presence of LiFSI, while the chemical shift of H in TMOS does not show a significant shift, suggesting the shielding effect originated from FB$_{135}$-FSI$^-$ interaction (Fig. 2g and Supplementary Fig. 13), which is consistent with the MD simulation results (Supplementary Figs. 14, 15 and Supplementary Table 8)[50].

To elucidate interaction-modulated Li metal reversibility, we analyze the electrode/electrolyte interfacial structure via constant charge MD simulations. The results reveal that anions are progressively expelled from the polarized electrode surface as repulsive electric field forces intensify (Fig. 3a). The number density distributions of ions and solvents (Fig. 3b and Supplementary Figs. 16, 17) show that FSI$^-$ anion expulsion still occurs in the LiFSI-2.5TMOS-3.0FB$_{123}$ and LiFSI-

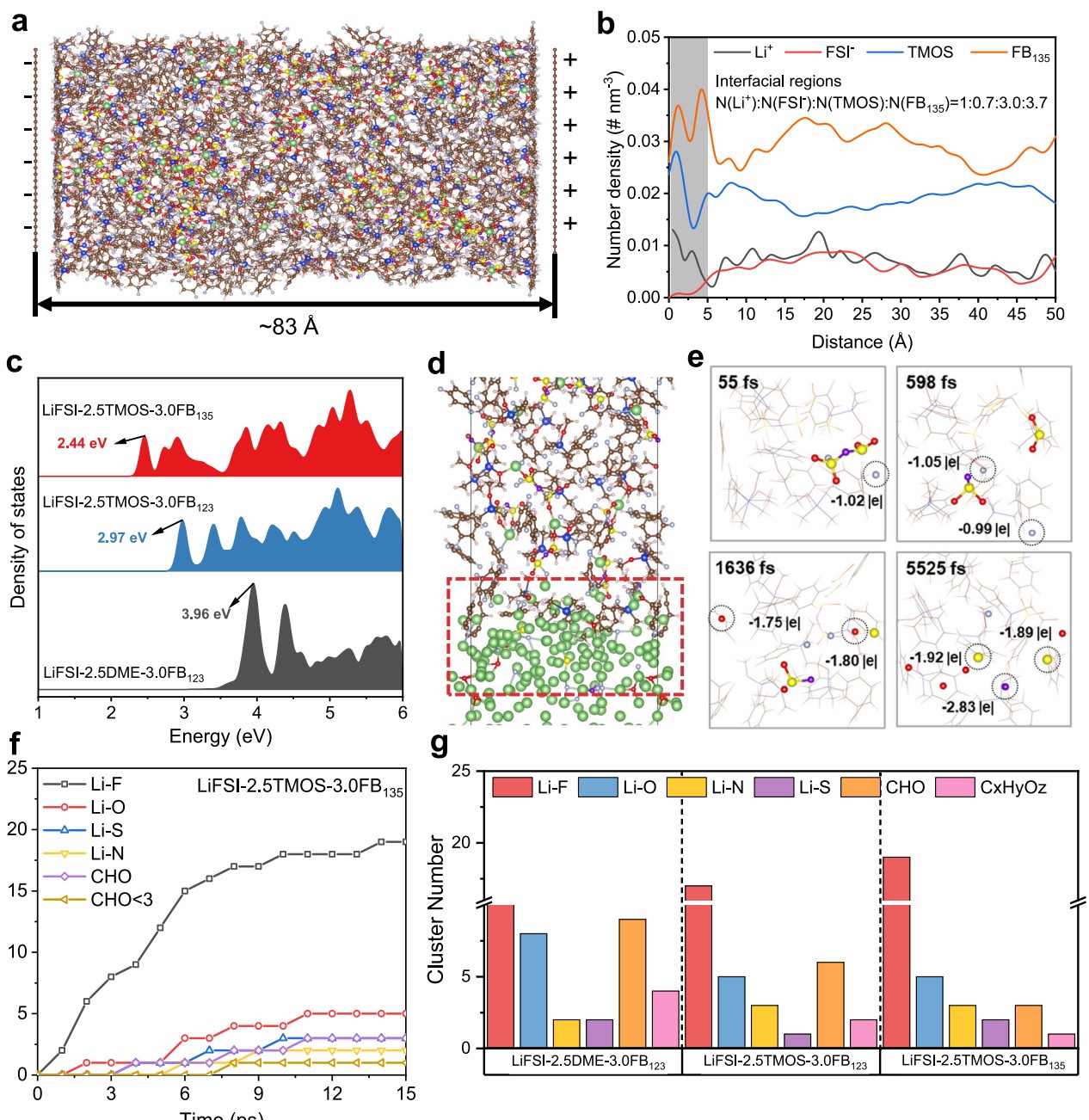

**Fig. 3 | The interfacial reaction simulations of various electrolytes. a** The snapshot of DEL structure of LiFSI-2.5TMOS-3.0FB$_{135}$. **b** Number density ($\rho$) profiles of ions and solvent molecules in LiFSI-2.5TMOS-3.0FB$_{135}$ electrolyte. **c** The projected density of states (PDOS) plots of FSI$^-$ in different electrolytes. **d** Snapshot revealing the chemical reactions at the Li metal-electrolyte interface in LiFSI-2.5TMOS-3.0FB$_{135}$ electrolyte. **e** Snapshots illustrating the time evolution of one FSI$^-$ anion in LiFSI-2.5TMOS-3.0FB$_{135}$ electrolyte. **f** Evolution of reaction products in LiFSI-2.5TMOS-3.0FB$_{135}$ electrolyte. **g** Statistical distribution of SEI species formed in different electrolytes. Color code: green: Li atom, red: O atom, yellow: S atom, purple: N atom, cyan: F atom, blue: Si atom, tan: C atom, white: H atom.

2.5TMOS-3.0FB$_{135}$ system, but the extent of this reduction is greatly mitigated due to the strong Li$^+$-FSI$^-$ pair interaction. We further evaluated the decomposition tendency of anion and solvent substances in different electrolytes using trajectory-average projected electronic density of states (PDOS)[51,52]. The reduction stability of molecules or anions directly correlates with their lowest unoccupied molecular orbital (LUMO) level, where a lower LUMO level indicates a higher likelihood of reduction[53] (Fig. 3c, Supplementary Fig. 18 and Supplementary Data 1). In the LiFSI-2.5TMOS-3.0FB$_{135}$ system, the FSI$^-$ anions tend to be reduced firstly due to the higher proportion of AGGs and longer pair lifetime, which is further supported by interfacial simulation using a Li/electrolyte model (Fig. 3d). Figure 3e demonstrates the rapid degradation of one FSI$^-$ anion into several inorganic fragments in the LiFSI-2.5TMOS-3.0FB$_{135}$ system. To clearly illustrate the variation in the SEI formation process, the evolution of reaction products during AIMD production was conducted (Fig. 3f, g and Supplementary Figs. 19–21). The high efficiency of LiF formation facilitates the deposition of Li in a granular and compact manner, leading to a high Li plating/stripping CE. Notably, the decomposition of TMOS is significantly inhibited throughout the entire simulation process (Supplementary Fig. 22), suggesting that the inorganic-rich SEI effectively prevents solvent decompositions on the Li-electrolyte interface.

## Li metal reversibility and battery performance

The compatibility between Li metal negative electrodes and electrolytes was evaluated using Li‖Cu cells according to modified Aurbach's cycling protocol[54,55]. CE reported process uncertainty of ±0.2%, as described in Supplementary Note 2[56]. The baseline electrolyte (BE, 1 M LiPF$_6$ EC/DMC) could not sustain such a measurement protocol due to the severe side reactions (Fig. 4a). In sharp contrast, LiFSI-2.5TMOS-3.0FB$_{135}$ electrolyte achieves a CE of approximately ~99.75%. The high CE achieved with LiFSI-2.5TMOS-3.0FB$_{135}$ indicates its ability to facilitate the rapid formation of protective SEI on Li metal negative electrode. The ratio of 1:2.5:3.0 was chosen to maximize the ionic conductivity (Supplementary Fig. 23). Furthermore, in the full plating/stripping tests with a fixed capacity of 3.0 mAh cm$^{-2}$ and a current density of 0.5 mA cm$^{-2}$, the Li‖Cu cells using LiFSI-2.5TMOS-3.0FB$_{135}$ reach the average CE of 99.6% and the stable cycle for over 3 months at a high areal capacity of 3 mAh cm$^{-2}$ (Fig. 4b). As the cycling progresses, the overpotential of cells with BE rapidly increases from 76 mV to 160 mV, which could be attributed to the side reactions triggered by the high areal capacity. On the contrary, the polarization of cells with LiFSI-2.5TMOS-3.0FB$_{135}$ remains relatively stable and is the lowest compared to other electrolytes (Supplementary Fig. 24). The stable Li‖Li plating/stripping over 2500 h (Supplementary Fig. 25) for all tested FB-based electrolytes demonstrates the long-term stability between the selected solvents (siloxane-, and FB-based solvents) and Li metal electrodes. Among them, LiFSI-2.5TMOS-3.0FB$_{135}$ exhibits high interfacial compatibility, as evidenced by the minimal evolution of overpotential. Additionally, the LiFSI-2.5TMOS-3.0FB$_{135}$ exhibit faster interfacial Li$^+$-transfer kinetics than those using LiFSI−2.5TMOS−3.0FB$_{123}$, as observed from exchange current density (0.24 mA cm$^{-2}$ for LiFSI-2.5TMOS-3.0FB$_{135}$ vs. 0.16 mA cm$^{-2}$ for LiFSI-2.5TMOS-3.0FB$_{123}$, Supplementary Fig. 26).

In addition to the favorable compatibility with Li metal, the oxidation stability of electrolytes is also crucial for the long-term cycling durability of high-energy full cells. Linear sweep voltammetry (LSV) curves reveal that the BE exhibits relatively poor anodic stability, evidenced by the sudden current increase in potential above 4.4 V (Supplementary Fig. 27). Conversely, both the LiFSI-2.5TMOS-3.0FB$_{135}$ and LiFSI-2.5TMOS-3.0FB$_{123}$ electrolytes present a high oxidation potential over 5.0 V. Due to the poor electrode/electrolyte interface stability, the 20 μm-Li‖LCO cells with BE rapidly deplete the active lithium, resulting in 80% CR within 50 cycles (Fig. 4c). In contrast, the other designed electrolytes significantly improve the cycling performance, achieving

80% CR after 187 (LiFSI-2.5TMOS-3.0FB$_{123}$), 251 (LiFSI-2.5TMOS-3.0FB$_{1245}$), and 345 (LiFSI-2.5TMOS-3.0FB$_{124}$) cycles, respectively. Notably, the LiFSI-2.5TMOS-3.0FB$_{135}$ exhibits the most stable cycling with a CR of 90% over 600 cycles. Such cycling stability still has advantages over the well-known LiFSI-1.2DME-3.0TTE LHCE system. The behavior was well supported by the extremely low overpotential (60.4 ± 9.4 mV, Supplementary Fig. 28) and interfacial resistance (29.4 Ω, Supplementary Fig. 29), while the cell impedance increment in LiFSI-2.5TMOS-3.0FB$_{123}$ is twice that of LiFSI-2.5TMOS-3.0FB$_{135}$ from the 50th cycle to the 300th cycle (Supplementary Fig. 30 and Supplementary Table 9). More importantly, the polarization during 450 cycles for the cell with LiFSI-2.5TMOS-3.0FB$_{135}$ was negligible compared to the reference electrolytes (Supplementary Fig. 31), indicating that surface layers have low resistance and high conductivity for Li$^+$. Even at 4.6 V, the Li‖LCO cells with LiFSI-2.5TMOS-3.0FB$_{135}$ deliver a CR of 93% over 200 cycles with a negligible voltage hysteresis (Supplementary Fig. 32). Significantly, the FB-based system also exhibits much better compatibility than BE for the Li‖4.4 V-NCM811 cells. Among them, the cells using LiFSI-2.5TMOS-3.0FB$_{135}$ retain a CR of 90% after 300 cycles (Supplementary Fig. 33), while the cells using BE show a continuous capacity decay with a retention of 80% after 80 cycles. Cu‖LiNi$_{0.5}$Co$_{0.2}$Mn$_{0.3}$O$_2$ (NCM523) pouch cells initially active material-free negative electrode were used to maximize energy density (Fig. 4d). Compared to other Li metal-friendly electrolytes, the pouch cell using the LiFSI-2.5TMOS-3.0FB$_{135}$ electrolyte achieves 76% CR over 100 cycles owing to favorable Li utilization under fluoride-rich surface layer protection.

## Electrode characterization and interphase chemistry

After illustrating the relationship between ion-solvent interactions and Li metal reversibility, we focused on the microstructural dimensions, mechanical attributes, and transport properties of the SEI. These aspects serve as pivotal links between microscopic interactions and practical efficiency, providing valuable insights into the intricate working fundamentals of these electrolyte systems. Compared to the noodle-like dendritic morphology of Li deposits in BE, both Li deposits from LiFSI-2.5TMOS-3.0FB$_{135}$ and LiFSI-2.5TMOS-3.0FB$_{123}$ electrolytes exhibit smooth morphology (Supplementary Fig. 34). Notably, Li metal negative electrode recovered from LiFSI-2.5TMOS-3.0FB$_{135}$ electrolyte exhibits a more flat and denser morphology, as evidenced by the statistical average thickness and corresponding standard deviation (20.2 ± 0.7 μm for LiFSI-2.5TMOS-3.0FB$_{135}$ vs. 25.7 ± 1.5 μm for LiFSI-2.5TMOS-3.0FB$_{123}$, Fig. 5a). The favorable Li deposition morphology achieved with LiFSI-2.5TMOS-3.0FB$_{135}$ greatly suppresses Li metal and electrolyte consumption, enabling higher Li plating/stripping CE.

The chemical composition and spatial distribution of as-formed SEI in various electrolytes were evaluated by using time-of-flight secondary-ion mass spectroscopy (ToF-SIMS) (Fig. 5b). Inorganic species (F$^-$, LiS$^-$, LiF$^-$) are homogeneously spread over the top surface of the Li metal negative electrode, and their signal peaks decay quickly after sputtering (Fig. 5c), which indicates the formation of LiF-rich and thin SEI layers in LiFSI-2.5TMOS-3.0FB$_{135}$. From the 3D maps presented in Supplementary Fig. 35, the signal of C$_6$H$_5^-$ was detected on the Li metal surface, indicating the decomposition of FB$_{135}$ and its participation in SEI formation. Notably, the accumulation of CHO$^-$, associated with solvent decomposition is negligible in the analyzed area. The in-depth X-ray photoelectron spectroscopy (XPS) of cycled Li metal further highlighted the surface characteristics in different electrolytes. As shown in Supplementary Fig. 36, the weak intensity of C-C/C-H (284.8 eV), C-O (286 eV), and ROCO$_2$ (288 eV) peaks[57] demonstrates that SEI layers formed in LiFSI-2.5TMOS-3.0FB$_{135}$ have relatively low organic content, distinguishing them from the SEI layers from reference electrolytes (Supplementary Fig. 37)[58]. Meanwhile, the peak associated with N (N-(SO$_x$)$_2$, 398.2 eV, N 1$s$) and S (SO$_2$F, 169 eV, SO$_x$, 166 eV, S$^{2-}$, 163.2 eV, S 2$p$) implies that the SEI in LiFSI-2.5TMOS-

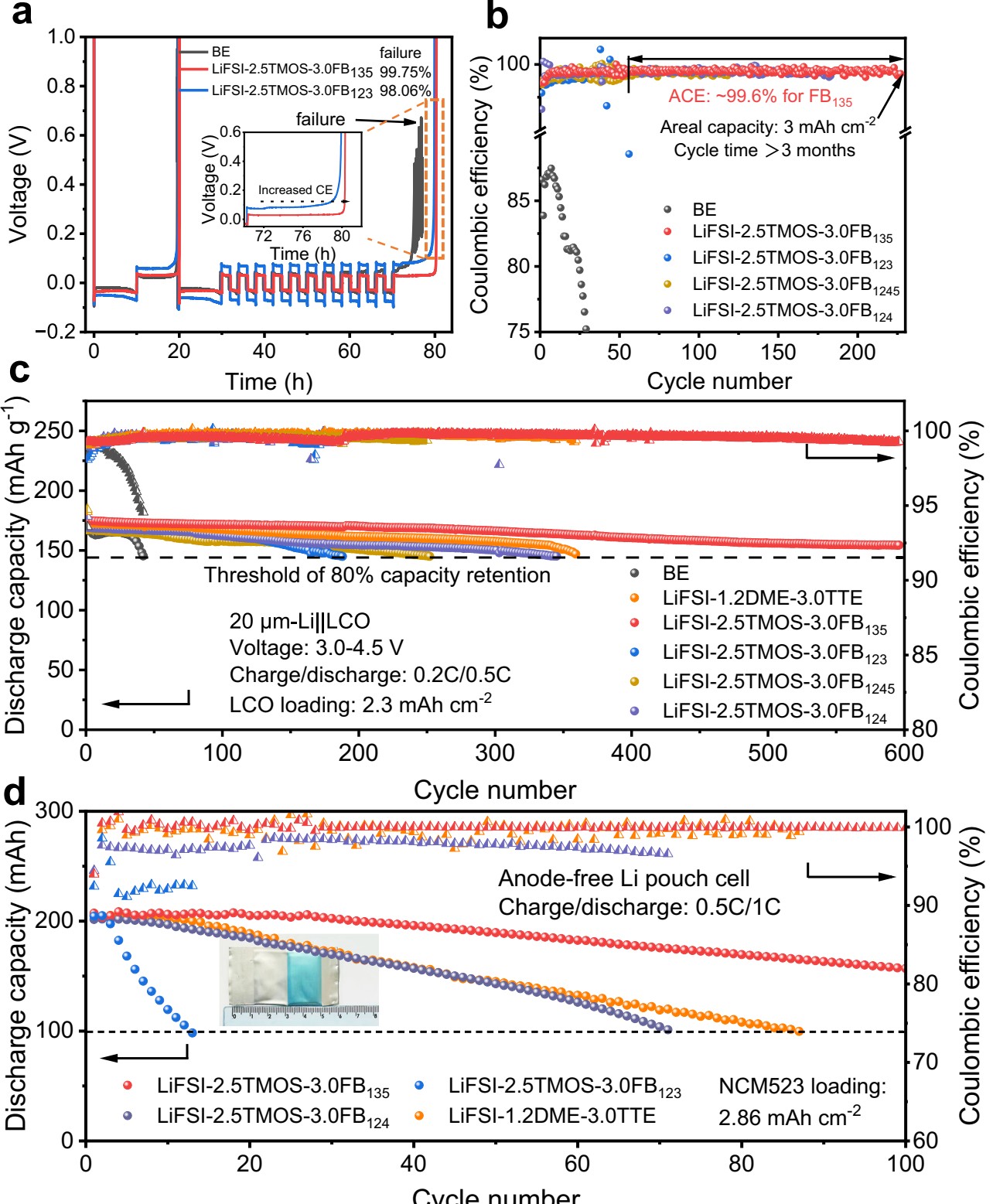

**Fig. 4 | Electrochemical performance of the designed electrolytes. a** Li metal CE in Li‖Cu cells for different electrolytes (0.5 mA cm⁻²) under 25 °C. **b** Long-term cycling stability of Li‖Cu cells for different electrolytes (0.5 mA cm⁻², 3 mAh cm⁻²) under 25 °C, ACE: average Coulombic efficiency. **c** Long-term cycling stability of 20 μm-Li‖4.5 V-LCO cells at 0.2 C/0.5 C charge/discharge (LCO loading: 2.3 mAh cm⁻², specific current: 1 C = 180 mA g⁻¹) under 25 °C. **d** Long-term cycling stability of Cu‖NCM523 pouch cells at 0.5 C/1 C charge/discharge (NCM523 loading: 2.86 mAh cm⁻¹, specific current: 1 C = 200 mA g⁻¹) under 25 °C.

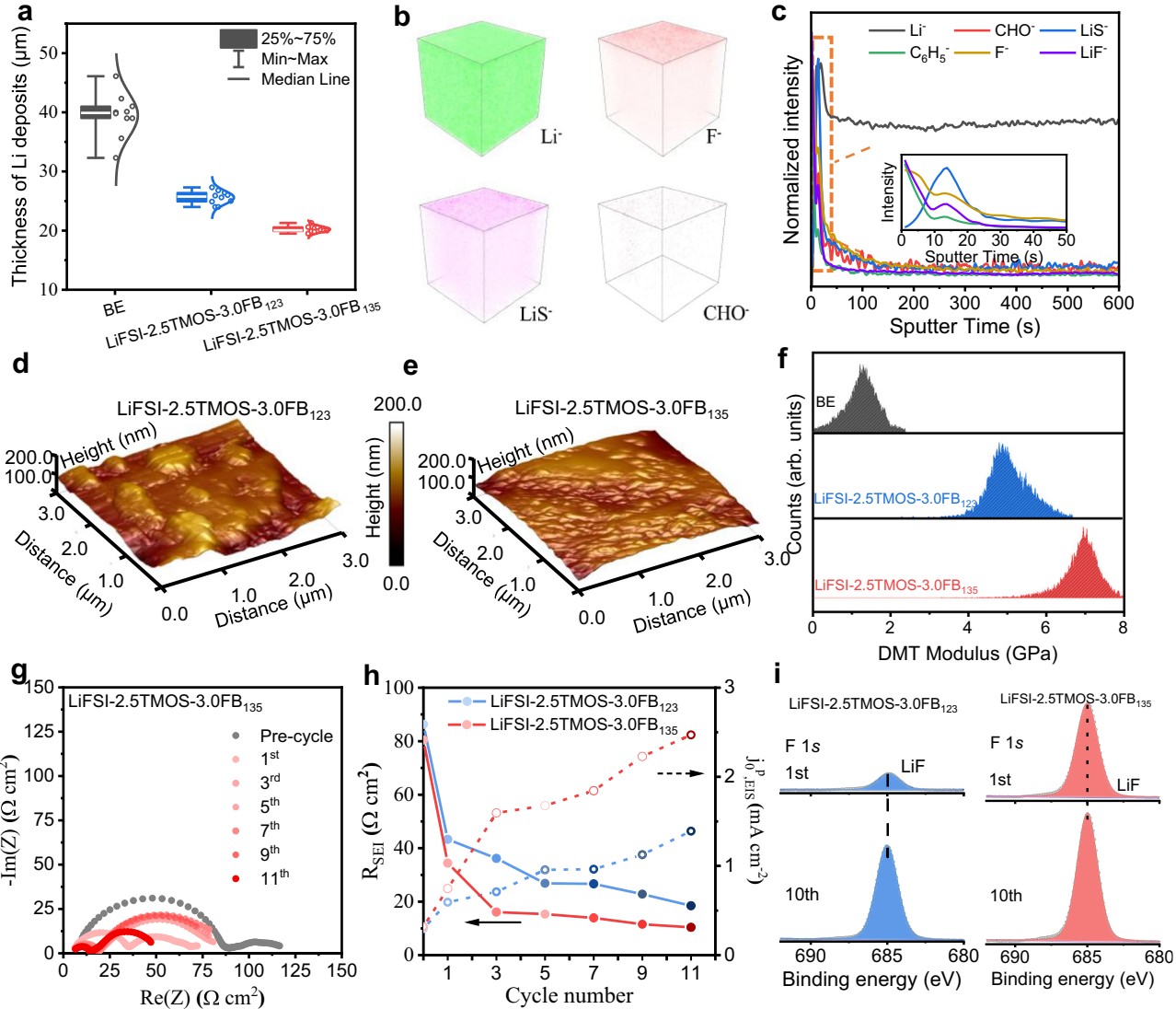

**Fig. 5 | Characterizations of Li metal/electrolyte interphase. a** Statistical comparison of Li deposit thickness on the cycled Li metal negative electrodes from Li||Cu cells after 20 cycles (0.5 mA cm², 4 mAh cm²) at 25 °C in different electrolytes. Data were measured from 10 Li layers across and are presented as box plots, where the top and bottom lines represent the maximum and minimum values, the top and bottom boundaries of the box represent the first quartile and the third quartile, the center line represents the mean, and the whiskers represent the fitted normal distribution curve. **b** ToF-SIMS 3D renders of several representative negative secondary ions (Li⁻, F⁻, LiS⁻, CHO⁻) on the cycled Li metal negative electrodes from Li||Cu cells after 20 cycles (0.5 mA cm², 4 mAh cm²) at 25 °C in LiFSI-2.5TMOS-3.0FB₁₃₅.

**c** Sputter time profiles showing normalized intensity of different fragments (Li⁻, CHO⁻, LiS⁻, C₆H₅⁻, F⁻, LiF⁻). AFM morphology at a region of 3.0 × 3.0 μm² on Li metal negative electrodes from Li||Cu cells after 20 cycles (0.5 mA cm², 4 mAh cm²) at 25 °C in (**d**) LiFSI-2.5TMOS-3.0FB₁₂₃ and (**e**) LiFSI-2.5TMOS-3.0FB₁₃₅. **f** Distribution of DMT modulus in different electrolytes. **g** Nyquist plots of Li||Li cells collected in LiFSI-2.5TMOS-3.0FB₁₃₅ after different Li plating and stripping cycles (0.5 mA cm⁻², 1 mAh cm⁻²) at 25 °C. **h** SEI resistance and pseudo-exchange current density for different electrolytes. **i** F 1s XPS spectra of cycled Li metal negative electrodes in different electrolytes after 1 and 10 cycles (0.5 mA cm², 4 mAh cm²) at 25 °C.

3.0FB₁₃₅ mainly originated from both FSI⁻ anion and FB₁₃₅ molecule. To quantify the LiF content in different electrolytes, the LiF-peak area ratio obtained from F 1s spectra (Supplementary Fig. 38). LiFSI-2.5TMOS-3.0FB₁₃₅ exhibits a higher LiF content ratio (~54%) compared to LiFSI-2.5TMOS-3.0FB₁₂₃ (~25%) and BE (~10%). This high LiF content in LiFSI-2.5TMOS-3.0FB₁₃₅ contributes to the strength and stiffness of the SEI layers.

Furthermore, the homogeneity and mechanical properties of the SEI were probed using the atomic force microscopy (AFM) technique[59]. In LiFSI-2.5TMOS-3.0FB₁₃₅ electrolyte, the SEI exhibited a maximum roughness of 17.5 nm (Fig. 5e). Comparatively, the SEI formed in other electrolytes exhibited higher maximum roughness values (e.g., 28.1 nm for LiFSI-2.5TMOS-3.0FB₁₂₃ and 127 nm for BE, Fig. 5d and Supplementary Fig. 39). More importantly, the Derjaguin-Müller-Toporov

(DMT) modulus of SEI formed in LiFSI-2.5TMOS-3.0FB₁₃₅ on the Li metal negative electrode is 7.1 GPa (Fig. 5f), which is much higher than that formed in LiFSI-2.5TMOS-3.0FB₁₂₃ (4.8 GPa). The comprehensive mechanical properties of LiFSI-2.5TMOS-3.0FB₁₃₅ validate that the dispersion-dominated electrolyte tends to prioritize the anion decomposition at the interface, thereby imparting robust interfacial chemistry and rigid bulk phase to Li deposits.

The transport properties and dynamic evolution of SEI were further measured using electrochemical impedance spectroscopy (EIS) in symmetric Li||Li cells[60]. As shown in the Nyquist plots in Fig. 5g and Supplementary Fig 40, Li plating/stripping consistently yields smaller impedance, attributed to breakage of the original surface film and formation of native SEI. After 10 cycles, substantial variations in $R_{SEI}$ magnitudes were observed among different electrolytes (Fig. 5h and

Supplementary Fig. 40c), BE displays the $R_{SEI}$ magnitude (710.1 Ω cm²), while the developed electrolytes demonstrated significantly lower values (18.9 Ω cm² LiFSI-2.5TMOS-3.0FB$_{123}$ and 10.4 Ω cm² for LiFSI-2.5TMOS-3.0FB$_{135}$). This notable reduction in interfacial resistance correlates with enhanced transport kinetics, as evidenced by the larger pseudo exchange current density $j^0_{0,EIS}$ of 2.5 mA cm⁻² in LiFSI-2.5TMOS-3.0FB$_{135}$. Moreover, the rapid dynamic stabilization process (only three cycles) of LiFSI-2.5TMOS-3.0FB$_{135}$ demonstrates its high efficiency of SEI formation, which can also be validated by analyzing the evolution of LiF content (Fig. 5i). Compared to the continuous increase of LiF in LiFSI-2.5TMOS-3.0FB$_{123}$ electrolyte, the LiF content remained relatively stable after 10 cycles in LiFSI-2.5TMOS-3.0FB$_{135}$, suggesting rapid defluorination and the formation of a compact LiF-rich SEI layer. Such inorganic-rich and larger-modulus SEI layer tends to have lower interfacial impedance and enhance the CE of lithium plating/stripping processes.

The serious side reactions and mechanical stress can lead to intragranular cracks within positive electrode particles, resulting in cell degradation during cycling[61,62]. SEM images reveal the presence of particle structure cracks for LCO cycled in LiFSI-2.5TMOS-3.0FB$_{123}$, while LiFSI-2.5TMOS-3.0FB$_{135}$ electrolyte effectively suppresses such cracks (Supplementary Fig. 41). With high-resolution transmission electron microscopy (HR-TEM) combined with energy dispersive X-ray analysis, a uniform and thin cathode electrolyte interphase (CEI) layer of ~5 nm is identified on LCO surface in LiFSI-2.5TMOS-3.0FB$_{135}$ (~13 nm for LiFSI-2.5TMOS-3.0FB$_{123}$ Fig. 6a, b and Supplementary Fig. 42), with the intact $R$-3m phase confirmed by the electron diffraction pattern (Fig. 6c). Detailed CEI compositions were examined by ToF-SIMS for cycled LCO in LiFSI-2.5TMOS-3.0FB$_{135}$. Inorganic species including $LiF_2^-$ and $LiS^-$ are found within the detected CEI layer, while the signal of $C_6H_5^-$ indicates the association of CEI film formation with FB$_{135}$ decomposition (Fig. 6d, e and Supplementary Fig. 43). To unveil the underlying mechanism, the F⁻ transferring behaviors of FB$_{135}$ and FB$_{123}$ were studied on charged LCO surface by density functional theory (DFT) calculations (Supplementary Data 2). It is found that the FB$_{135}$ molecules possess low defluorination energy (0.15 eV) compared to FB$_{123}$ molecules (1.45 eV), suggesting inorganic CEI layers that effectively passivate the aggressive LCO surface is favorable in the LiFSI-2.5TMOS-3.0FB$_{135}$ system (Fig. 6f and Supplementary Figs. 44, 45).

Based on the above theoretical predictions and experimental evidence, we conclude that the enhanced performance of our electrolytes is primarily attributed to the modulation of solvent-FSI⁻-diluent interactions. This design strategy enables favorable electrode-electrolyte interface formation and high CE (Fig. 6g). Moreover, the quantitative determination of solvent-ion interaction parameters enables precise characterization of the dynamic coordination equilibrium between anions and cations, along with their time-dependent characteristics. This mechanistic insight provides a critical fundamental understanding of the structure-property relationships governing ionic transport phenomena, particularly in elucidating the quantitative dependence of macroscopic electrochemical parameters (such as ionic conductivity and Li⁺ transference number) on the molecular-scale solvation dynamics.

In summary, we propose the quantitative exploration of NCI interactions (electrostatic, dispersion, and induction) for understanding the electrolyte properties by defining the $E_{ele}/E_{ind}$ (Li⁺) and $E_{ele}/E_{dis}$ (FSI⁻) interaction motif descriptors. The ideal electrolyte for LMBs should not only incorporate at least one solvent capable of effectively solvating Li⁺ (with an $E_{ele}/E_{ind}$ (Li⁺) > 1.3) but also demonstrate well-balanced interactions among its components to prevent the phase transition/separation. Additionally, it should exhibit minimal interactions in ion-solvent pairs (with an $E_{ele}/E_{dis}$ (FSI⁻) < 1.0) to facilitate the kinetics of Faradaic reactions and mitigate the undesired side interfacial reactions at the electrode surface. Based on the electrolyte design principle, we report a representative electrolyte, *i.d.*, LiFSI

dissolved in TMOS and FB$_{135}$, where both TMOS and FB$_{135}$ predominantly interact with FSI⁻ through dispersion forces rather than electrostatic forces. As a result, the designed LiFSI-2.5TMOS-3.0FB$_{135}$ demonstrates a CE of ~99.7% (±0.2%) and exhibits long-term cycling over 3 months in full platting/stripping test with a high capacity of 3.0 mAh cm⁻². The 20 μm-Li||2.3mAh·cm⁻² LCO cells with LiFSI-2.5TMOS-3.0FB$_{135}$ retain >90% CR after 600 cycles, far exceeding the BE (80% CR after 50 cycles). Remarkably, Cu||NCM523 pouch cells with LiFSI-2.5TMOS-3.0FB$_{135}$ achieve stable cycling over 100 cycles. This work demonstrates a feasible electrolyte design principle at the molecule-interaction level, providing valuable insights for the development of next-generation LMBs with high reversibility.

## Methods
### Materials
LiCoO$_2$ (LCO, 99.5%) powder, LiNi$_{0.8}$Co$_{0.1}$Mn$_{0.1}$ (NCM811, 99.5%) powder and conductive carbon (Super C45, 99.5%) were purchased from Hefei Keijing Co., Ltd. 450-μm-thick and 20-μm-thick lithium chips (on Cu foil) were supplied by Tianjin China Energy Lithium Co., Ltd. N-Methyl pyrrolidone (NMP, 99.5%) and Polyvinylidene fluoride (PVDF, molecular weight: ~50 W, 99.5%) was obtained from Duoduo Co., Ltd. Lithium bis(fluorosulfonyl)azanide (LiFSI, 99%), ethylene carbonate (EC, 99%) and dimethyl carbonate (DMC, 99%) were provided by Changde Dadu New Material Co., Ltd. 1,2-Dimethoxyethane (DME, 99%), trimethyl methoxysilane (TMOS, 99%), 1,2,4-trifluorobenzene (FB$_{124}$, 99%), 1,2,3-trifluorobenzene (FB$_{123}$, 99%), 1,2,4,5-tetra-fluorobenzene (FB$_{1245}$, 99%), 1,3,5-trifluorobenzene (FB$_{135}$, 99%), 2-Methyltetrahydrofuran (Me-THF, 99%), tert-Butyl methyl ether (TBME, 99%) 1h-perfluorohexane (CFH, 99%), Methyl perfluorobutyl ether (MNE, 99%), and 1,1,1,2,2-Pentafluoro-3-methoxypropane (PMP, 99%) were purchased from Shanghai Meryer Biochemical Technology Co., Ltd.

### Preparation of electrolyte and electrode
The electrolytes were prepared by dissolving LiFSI in different solvent mixtures. The molar ratios of LiFSI to solvents were used to label different electrolytes. For example, the notation LiFSI-2.5TMOS-3.0FB$_{135}$ signifies that the mole ratios of LiFSI, TMOS, and FB$_{135}$ in the electrolyte are 1:2.5:3.0. All solvents were dried with activated 4 Å molecular sieves before use. The reference electrolytes are 1 M LiPF$_6$ EC/DMC (1:1 v/v, named BE electrolyte). The LCO positive electrodes were obtained by casting the mixture (composed of 96 wt.% LCO, 2 wt.% Super C45, and 2 wt.% PVDF, with an areal loading of 2.3 mAh cm⁻²) onto the Al foil. For NCM811 positive electrodes, the mixture of 2 wt.% Super C45, 2 wt.% PVDF and 96 wt.% NCM811 as active material (with an areal loading of 2.3 mAh cm⁻²) was coated in an Al foil and the electrodes were further dried at 80 °C overnight under vacuum. The solvent-to-solid ratio in the slurry is ~1.0 mL g⁻¹. The active material loading of NCM811 and LCO was ~13 mg cm⁻². A 20-μm thick lithium metal is cut into small discs with a diameter of 14 mm to serve as the negative electrode. Celgard 2325 separator (Thickness: 25 μm; Lateral dimension: 100 mm; Porosity: 40%; Average pore size: 0.028 μm) is cut into small discs with a diameter of 19 mm. The single-side coated positive electrode for the coin cell was cut into circular pieces with a diameter of 12 mm, and each coin cell used 70 μL electrolyte. All electrolytes and electrodes were stored inside an argon-filled glove box, where the moisture and oxygen content were less than 0.1 ppm. Industry-level Cu||NCM523 pouch cells (NCM523 Loading: 18.3 mg cm⁻², total weight: 2.3 g, Capacity: 200 mAh) were purchased from LI-FUN Technology.

### Electrochemical tests
CR2025 coin cells (Guangdong Canrd New Energy Technology Co.) were electrochemically evaluated at 25 °C using a climate-controlled chamber (BLC-300, ShangHai BOLAB). The charge-discharge cycling performance tests were performed on a LAND system (Wuhan LAND

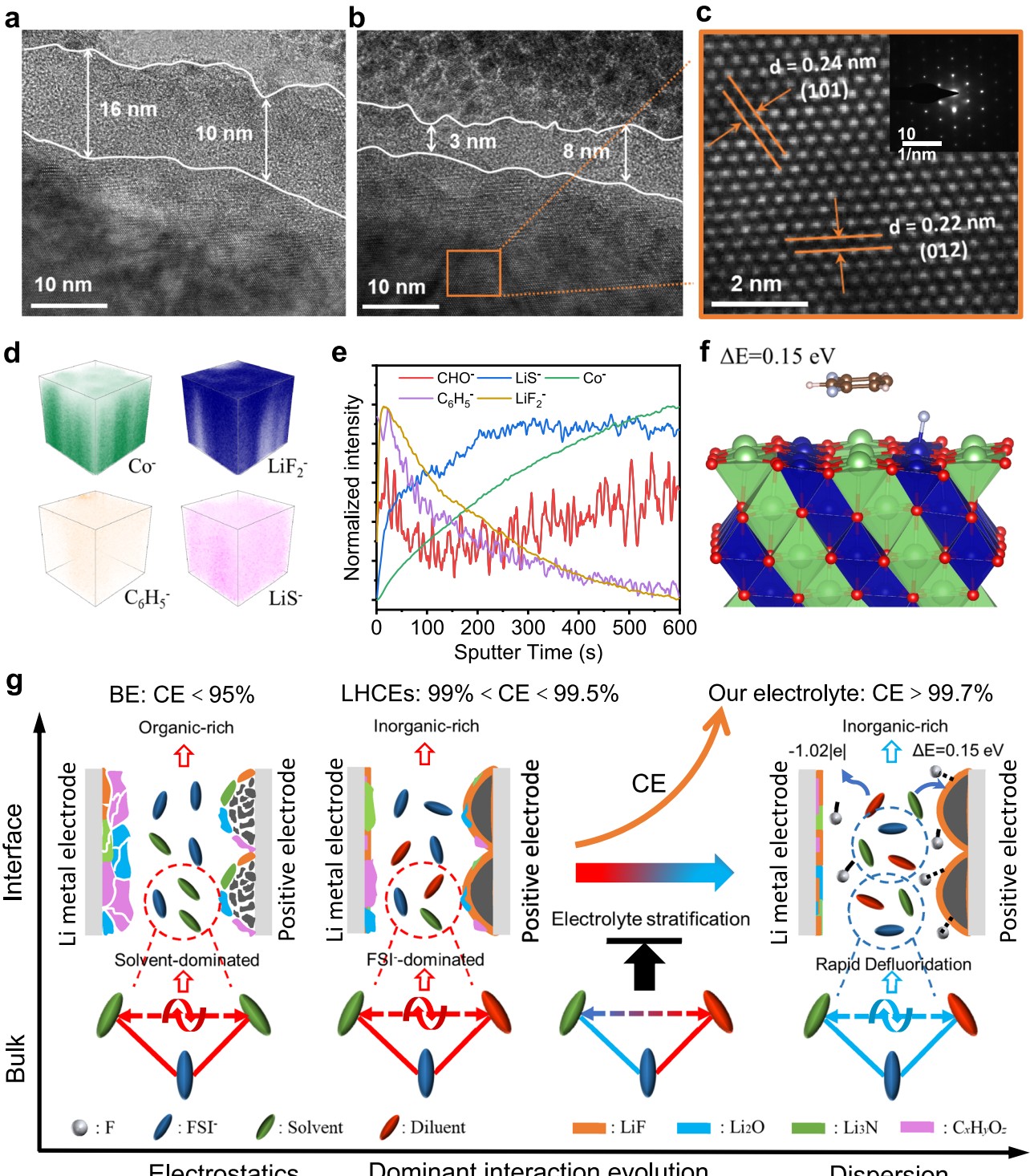

**Fig. 6 | Characterizations of positive electrode/electrolyte interphase, and interaction-regulated design for high-CE electrolyte.** HR-TEM images of cycled LCO positive electrode in a fully discharged state after 200 cycles (0.2 C/0.5 C charge/discharge) at 25 °C using LiFSI-2.5TMOS-3.0FB$_{123}$ (**a**) and LiFSI-2.5TMOS-3.0FB$_{135}$ (**b**), the enlargement for HR-TEM images and corresponding selected area electron diffraction (SAED) pattern (**c**). ToF-SIMS date for the cycled LCO positive electrode in a fully discharged state after 200 cycles (0.2 C/0.5 C charge/discharge) at 25 °C in LiFSI-2.5TMOS-3.0FB$_{135}$, 3D maps (**d**) presenting the distribution of different fragments (Co$^-$, LiF$_2^-$, C$_6$H$_5^-$, LiS$^-$), Sputter time profiles (**e**) showing the normalized intensity of different fragments (CHO$^-$, LiS$^-$, Co$^-$, C$_6$H$_5^-$, LiF$_2^-$). **f** Oxidation reaction energy $\Delta E$ of F$^-$ transfer for FB$_{135}$ on the delithiated Li$_{36}$Co$_{48}$O$_{96}$ surface. **g** Relationship between interfacial interaction evolution and electrolyte properties. −1.02|e| and 0.15 eV are transferred electrons for LiF formation on Li metal and F$^-$ transfer energy on LCO in LiFSI-2.5TMOS-3.0FB$_{135}$, respectively. Color code: green: Li atom, dark blue: Co atom, red: O atom, cyan: F atom, tan: C atom, white: H atom.

Electronics Co., Ltd). For Li∥Cu Cell, the average CE was calculated as the ratio of the total stripping capacity to the deposited capacity. Aurbach's CE protocol: Deposit 5 mAh cm⁻² Li on Cu (0.5 mA cm⁻²), strip to 1.0 V, then perform 10 cycles of 1 mAh cm⁻² stripping/plating. Finally, the remaining Li was stripped to 1.0 V. For Li∥4.5 V LCO and Li∥ 4.4 V NCM811 cells, the first three activation cycles were at 0.1 C, and then were at 0.2 C/0.5 C charge/discharge for long-term cycles. For Li∥ 4.6 V LCO cells, the first three activation cycles were at 0.1 C, and then were at 0.2 C for long-term cycles. LSV (2.0-6.0 V, 1.0 mV s⁻¹) and EIS (0.1 Hz-100 kHz, 10 mV amplitude) were conducted using an Ivium-n-stat analyzer. EIS measurements included 12 data points per frequency decade after 30 s open-circuit stabilization. For the tests of pouch cells, all cells proceeded with a two-cycle formation at 0.1 C and degassed before cycling tests. The stack pressure for pouch cells was ~100 kPa.

## Characterization

Raman spectra of different electrolytes were measured by LabRAM HR Evolution with a 532 nm laser. In the nuclear magnetic resonance (NMR) tests, a coaxial internal insert that contained a standard NMR solvent (i.e., 0.1 M LiFSI in 2 vol % $H_2O$ + 98 vol % $D_2O$ as the reference) was inserted into the NMR tube to analyze the pristine microstructure of electrolyte by a Bruker AVIII 400 MHz Liquid NMR spectrometer.

Scanning electron microscopy (SEM) was obtained from a FEI Strata 400S microscope (Hitachi, Japan) at an accelerating voltage of 2 kV. Transmission electron microscopy (TEM) measurements were performed on a FEI Tecnai F20 microscope instrument at an accelerating voltage of 200 kV and EDS energy range of 0.1-30 keV. The Li metal negative electrodes and LCO positive electrodes were collected from the cycled cells that were disassembled and stored in the glovebox before characterizing.

X-ray photoelectron spectroscopy (XPS) tests were carried out on a scanning X-ray microprobe (Thermo Scientific ESCALAB 250Xi, with a monochromatic Al Kα X-ray source). The samples were etched by Ar⁺ ions (2 kV, 2 μA, 45° incident angle) with increasing the sputtering time (0 s, 120 s, 300 s, and 600 s) before the measurements. The XPS spectra are calibrated by using C 1$s$ (284.8 eV) as the reference peak. The cycled positive and negative electrodes were rinsed in 1 mL of the corresponding fresh TMOS solvent 3 times during the sample preparation.

Time-of-flight secondary ion mass spectrometry (ToF-SIMS, TOF.SIMS5-100) was operated with Cs⁺ sputtering beam (1 keV) over an area of 50 μm × 50 μm, and the current for the sputtering beam was steady and constant (60 nA). The roughness and mechanical properties of Li deposits were tested on Dimension Icon Atomic Force Microscope (AFM). The typical laser spot size is 3.0 × 3.0 μm² at the sample. Samples for ToF-SIMS and AFM analysis were prepared by plating Li onto Cu substrates (0.5 mA cm², 4 mAh cm⁻²) in Li∥Cu cells. All Li∥Cu cells for sample preparation were disassembled and washed in a glove box under an argon atmosphere. Then the samples were transferred to the stage of the spectrometer inside the glove box.

## Theoretical calculations

**Quantum chemistry calculations.** All calculations were performed using Gaussian 16 (A.03) and the xtb package (v6.6.0). MD simulations of Li⁺-solvent and FSI⁻-solvent complexes were conducted in a periodic cubic box (size determined by density), with stable configurations sampled from 200-ps trajectories. Initial geometries of dimer complexes extracted from MD trajectories were optimized via the GFN2-xTB method with GBSA solvation, followed by DFT refinement at the B3LYP-D3(BJ)/6-311+G(d, p) level, and frequency analysis confirmed the absence of imaginary frequencies[63]. Single-point energy calculations were then performed using CCSD(T)/jul-cc-pVTZ with Boys-Bernardi counterpoise correction to address basis set superposition errors (BSSE). Deformation energies were excluded to focus on intrinsic intermolecular interactions. Binding free energies were derived by combining the intrinsic interaction energy ($E_{int}$) and

thermal corrections ($\Delta\Delta G^{coor}$), as shown in Eq. 1:

$$G_{bind} = E_{int} + \Delta\Delta G^{coor} \tag{1}$$

Where the binding free energy correction ($\Delta\Delta G^{corr}$) to interaction to account for the temperature effect using the Eq. 2:

$$\Delta\Delta G^{coor} = \Delta G^{coor}_{com} - \Delta G^{coor}_{fra} \tag{2}$$

where $\Delta G^{coor}_{com}$ and $\Delta G^{coor}_{fra}$ represent the thermal corrections to the Gibbs free energy for the complex and the fragment, calculated at their corresponding equilibrium geometries, respectively.

Symmetry-adapted perturbation theory (SAPT) analyses were performed using the PSI4 program at the SAPT2+/aug-cc-pVDZ level (denoted as the 'silver' level), which provides high-accuracy binding energy (BE) values without requiring BSSE corrections due to SAPT's inherent insensitivity to basis set superposition errors[64]. Asymptotic behavior corrections and time-dependent DFT were excluded in this framework.

Reduction potentials were calculated via Eq. 3,

$$E_{reduction} = -[\Delta G_e + \Delta G_s(M^-) - \Delta G_s(M)]/F - 1.4 \tag{3}$$

where $\Delta G_e$ is the gas-phase electron affinity at 298.15 K, $\Delta G_s(M^-)$ and $\Delta G_s(M)$ represent the solvation free energies of the reduced and initial complexes, respectively, and $F$ is the Faraday constant.

**Classical molecular dynamics (cMD) simulations.** Classical molecular dynamics simulations were performed in the LAMMPS using the all-atom optimized potentials for liquid simulations (OPLS-AA) force-field, where the Li⁺ and FSI⁻ were described from previous publications[28,65]. Given the critical importance of cation-anion coordination in solvation structures, rigorous testing or the utilization of the systematic Molecular Dynamics Electronic Continuum (MDEC) model becomes imperative. To accurately capture ion-ion and ion-dipole interactions, charges for Li⁺ and FSI⁻ from the salts were appropriately scaled. In terms of the density, as well as Li⁺ coordination numbers within different electrolytes, the scaling factor of 0.70 for DME system and 0.75 for TMOS system demonstrates closer agreement with experiments and/or simulation results based on polarizable force field or density functional theory, as discussed in Supplementary Fig. 6. Before simulating, the electrolyte systems were constructed by placing the solvents, Li⁺, and anions based on their molar ratio and density information using Moltemplate (http://www.moltemplate.org/). In the simulation process, each electrolyte system initially performed an energy minimization at 0 K to obtain the ground-state structures until energy and force accuracy reached 10⁻⁴ and 10⁻⁶. Then, the electrolyte was heated to 350 K with a constant volume of 0.2 ns using a Langevin thermostat with a damping parameter of 100 ps. To get the structure of the electrolyte, NPT (1 bar, and 350 K) runs were first performed at 350 K for 5 ns and then 298 K for 5 ns to ensure the equilibrium salt dissociation. Then, 20 ns long NVT (298 K) runs were conducted and the last 5 ns were used to obtain the structure of the electrolyte. All simulations were run with a time step of 1.0 fs. Temperature and pressure were maintained using a Nosé-Hoover thermostat and barostat as defined in LAMMPS. The particle mesh Ewald method was used to calculate electrostatic interactions, with a real space cut-off of 1.2 nm and a Fourier spacing of 0.12 nm. A cut-off of 1.2 nm was used for non-bonded Lennard-Jones interactions. To further validate the MD results, polarizable force fields based on the classical Drude oscillator model (CDO) were also employed to study the solvation structure. The polarizable MD was performed by using LAMMPS with the help of DLPGEN package[5]. The statistics of solvation configuration were obtained by the Visual Molecular Dynamics (VMD) software[66].

Snapshots of the most probable solvation structures were also sampled from the simulation trajectory using VESTA software[67].

As for the electric double-layer (EDL) simulations, two graphene slabs were optimized via density functional theory (DFT) calculations in advance. The corresponding electrolyte systems were constructed to fit the lattice constant of the graphene slab. Then, the electrolyte system and graphene slabs were combined, and two graphene slabs were added to different net charges which stand for the positive and negative electrodes, respectively. This method of constant charge calculation accurately shows the direct relationship between changes in charge quantity and voltage fluctuations, allowing for a superior illustration of the exact distribution of solvation structures at varying voltage conditions. To reduce the Coulomb effect between the mirrored slabs owing to the periodic boundary condition in the z orientation, a vacuum layer of 20 Å thickness was added to the z-axis. Finally, the energy minimization and heat equilibrium processes were conducted in the NVT ensemble as above mentioned. The total simulation time was set as 5 ns at 300 K.

**Ab initio molecular dynamics simulations.** Electrolyte reduction behavior was investigated using AIMD simulations implemented in the Vienna Ab Initio Simulation Package (VASP)[68]. Calculations employed the Perdew-Burke-Ernzerhof-generalized gradient approximation (PBE-GGA) functional with DFT-D3 dispersion correction, projector augmented wave (PAW) pseudopotentials, and a 450-eV plane-wave cutoff. Initial LiFSI/solvent/diluent configurations (molar ratios matching experimental densities) were pre-optimized via molecular mechanics. Simulations used $\Gamma$-point sampling, a 1.0 fs timestep (tritium mass for protons), and 10 ps trajectories. pDOS of both anions and solvents were averaged over five configurations (2 ps intervals) for ensemble statistics.

For Li-metal/electrolyte interfacial reactions, Born-Oppenheimer AIMD simulations were conducted in CP2K/Quickstep using a hybrid Gaussian/plane-wave scheme[69]. Valence electrons were treated using the DZVP-MOLOPT-GTH basis sets, while core-electron interactions were described by Goedecker-Teter-Hutter (GTH) pseudopotentials. The Perdew-Burke-Ernzerhof (PBE) functional with Grimme D3 dispersion correction was employed for exchange-correlation effects. Periodic boundary conditions were implemented with a 400 Ry plane-wave cutoff for charge density expansion. Simulations used $\Gamma$-point sampling, a 1.0 fs timestep, and an electronic energy convergence criterion of $1 \times 10^{-5}$ eV. A 16-layer Li (100) slab (dimensions: $21.67 \times 18.39 \times 64.00$ Å$^3$) was constructed with the middle three layers fixed to mimic bulk behavior. A ~ 40 Å vacuum layer along the z-axis accommodated the electrolyte systems. LiFSI-2.5TMOS-3.0FB$_{135}$ (15 Li$^+$, 15 FSI$^-$, 30 TMOS, and 45 FB$_{135}$ molecules), LiFSI-2.5TMOS-3.0FB$_{123}$ (15 Li$^+$, 15 FSI$^-$, 30 TMOS, and 45 FB$_{123}$ molecules) and LiFSI-2.5DME-3.0FB$_{123}$ (15 Li$^+$, 15 FSI$^-$, 30 DME, and 45 FB$_{123}$ molecules) electrolytes were tested. Initial electrolyte configurations were generated via PACKMOL, followed by geometry quenching using density functional forces. AIMD trajectories were propagated at 350 K (Nose-Hoover thermostat) for 15 ps to analyze interfacial reactions.

**Periodic calculations.** Surface H and F transfer reactions were investigated using the VASP code with the spin-polarized PAW method and PBE-GGA functional. A 9-layer LiCoO$_2$ (104) slab model with 15 Å along the z-axis was constructed. The DFT + U was employed, where $U_{eff} = U-J$ ($U = 3.32$ eV, $J = 0$ eV) was applied on the Co d states[70]. Solvent molecules were adsorbed on the surface with four upper layers relaxed, while five lower layers remained fixed (energy/force convergence: $1 \times 10^{-5}$ eV/0.01 eV Å$^{-1}$). Calculations used $\Gamma$-point sampling, 500 eV plane-wave cutoff. Dipole correction was applied on the z axis.

## Data availability

All data that support the findings of this study are presented in the manuscript and Supplementary Information, or are available from the corresponding author upon request. Source data are provided with this paper.

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

## Acknowledgements

This work was supported by the Key R&D Program of Zhejiang (2023C01128), National Natural Science Foundation of China (22072134, 22161142017, and U21A2081), National Key Research and Development Program of China (2022YFE0206300), Natural Science Foundation of Zhejiang Province (LZ21B030002 and LR23B030002), and "Hundred Talents Program" of Zhejiang University.

## Author contributions

R.L., X.H. and X.F. conceived the idea and designed the research. R.L., X.H., H.Z. and J.W. performed electrochemical measurements. Y.F. and Y.S. performed the ToF-SIMS measurement. J.L. helped measure the AFM measurement. R.L., X.H., J.W., Y.H., M.Y. and T.D. analyzed data. R.L. conducted the theoretical calculations. Y.Y., X.X., Y.T., H.B.W. and L.F. participated in the discussions. R.L. and X.H. wrote and revised the original paper, and M.Y., L.C., Y.S., T.D. and X.F. edited and polished the paper.

## Competing interests

The authors declare no competing interests.
