## [Transparent Peer Review file · Nature Communications]

A path towards high lithium-metal electrode coulombic efficiency based on electrolyte interaction motif descriptor

Corresponding Author: Professor Xiulin Fan

Version 0:

Reviewer comments:

Reviewer #1

(Remarks to the Author)

In this work, the authors demonstrate a new indicator, the fraction of dispersion interaction, for localized high-concentration electrolyte design. Moreover, the electrochemical performance of their formulated LiFSI-2.5TMOS-3.0FB135 surpassed most reported electrolytes in high-voltage Li metal batteries with an even more aggressive test protocol. The manuscript presents a well-formulated fundamental principle/mechanism, extensive theoretical support illustrating the mechanism, and proof-of-concept validation through experiments. In summary, this reviewer believes that this manuscript provides valuable information for a wide range of readers of Nature Communications and revisions are suggested below.

1. Is the simulation system information in Table S6 in SI the same as in Figure 2? The simulation box depicted in Figures 2g and 2j is not cubic, yet Table S6 only provides the box length for one dimension. Please include the box lengths for all dimensions for clarity.
2. In Figure 2b, the authors visualized the cationic solvation species of various electrolytes to compare the size of the solvation clusters. However, the author did not give the corresponding evaluation method, mean or median? Discussion on possible errors is also needed.
3. On page 16, the electrolyte LiFSI-2.5TMOS-3.0FB135 exhibits faster interfacial Li⁺-transfer kinetics than those using LiFSI-2.5TMOS-3.0FB123. Could the authors provide a more detailed discussion of the reasoning behind this?
4. In Figures S11 and S12, the temperature and energy curves show a trend of increasing first and then decreasing. In particular, in the time range of 4~8 ns, the temperature is about 500 K. The parameters used in the simulation need to be provided to better understand such a situation. In addition, what are the thermostat and the barostat in the CMD simulations with the OPLS force field?
5. The ex-situ XPS measurements suggest a higher content of inorganic SEI/CEI for LiFSI-2.5TMOS-3.0FB135 electrolytes. Please provide details regarding sample preparation and/or transfer, which are thought to have a huge impact on the XPS results.

Reviewer #2

(Remarks to the Author)

The authors report a new criteria for electrolyte based on the ratio of induction-based and Coulombic interaction energies between Li⁺ and solvent components and the ratio of dispersion-based and Coulombic interaction energies between the solvent and FSI⁻. In using these calculated ratios, the authors go onto design a solvent/diluent system (methoxytrimethylsilane/fluorobenzene with various fluorination sites) that delivers an extremely high cycling efficiency for Li metal. While the authors claim that these quantities represent a fundamentally new design criteria for Li metal battery electrolytes, I find that the physical description of these criteria to be relatively vague, with little evidence of its differentiation from previous design criteria. Additionally, the application of this criteria in this work, which is ultimately correlative to performance in a relatively narrow and curated system set, casts further doubt on its potential utility to the field. Despite the impressive claimed performance and the clear need for more sophisticated electrolyte design criteria, I cannot recommend that this article be published in its current form. Addressing the following points may improve this.

- The E_{ele}/E_{ind} (Li⁺) and E_{ele}/E_{disp} (FSI⁻) design criteria seems to have little rooting in the physical phenomena

corresponding to Li metal reversibility. For instance, the authors claim that these interaction energy ratios inherently define solvation structure, ionic conductivity, and SEI composition, but do not seem to make a concerted effort to demonstrate that this is the case. Instead, they demonstrate that these quantities can be correlated to the observed Coulombic Efficiency of a relatively narrow range of electrolytes. For the proposed DFT-based design criteria to be actually useful, I would heavily recommend that the physical origin of the Coulombic efficiency be at least hypothesized. Otherwise, the usefulness of the E/E ratios in predicting future systems is hazy at best.

- The authors criticize, perhaps correctly, the field's reliance on "strong" and "weak" solvation as system descriptors in the introduction. However, it seems as though the author's the deepest discussion of the Eele/Eind (Li+) and Eele/Edisp (FSI-) quantities revolve around the notion that low ratios represent weak overall interactions. If this is the case, how are these interaction energy ratios any more useful than the "strong" and "weak" classifications currently used in the field?
- A rigorous comparison between the E/E ratios and the measured solvation energy (J. Am. Chem. Soc., 2021, 143, 10301–10308.) would be extremely valuable to demonstrate whether these quantities actually yield insights that solvation energy alone does not.
- Along these lines, the ways that "Ratio of Dispersion" (x-axis in figure 1c) is different than weakness of solvation is unclear and should be discussed at length with precise language.
- The authors claim that the interaction energy ratios in part directly predict electrolyte phase separation and salt precipitation seems to have little to no support in the data, save for one instance in which a single system was observed to display phase separation (LiFSI/DME/FB135). Moreover, it is difficult to believe that the miscibility of 2 solvents, even in the presence of salt, is unrelated to the interaction energies between the diluent and solvent. It is recommended that claims regarding the ability of the DFT energy design criteria to predict things such as multi-component phase miscibility, which is itself highly entropic in nature, be removed from the work given the lack of evidence and data.
- Similarly, the cynical reading of the E/E ratios role in dictating the calculated solvation structures within the FBxyz series does not appear to be any different than their respective solvation energies towards lithium, which would similarly describe the aggregate sizes and compositions. The correlation between such aggregation and CE is also well-established.
- Critical information from the cMD calculation methods are missing, including box sizes, number of atoms/molecules, periodicity (the renderings shown in figure 2 appear to be rectangular?), damping parameters for NPT/NVT steps, minimization convergence criteria, etc.
- A CE of 99.9 % approaches the regimes at which the precision of the battery cyclers used begins to significantly affect the measured CE. It is heavily recommended that the authors measure the delivered current of the machine with a high precision probe relative to the current that the machine claims to deliver as a way to measure the accuracy of the reported CE value. For example, in other highly reversible systems such error has been shown to be on the order approaching +/- 0.30 % (Y. Yang, et al. Joule, 2019, 3, 1986–2000.)
- The baseline electrolyte used in the electrochemical performance evaluation is known to show extremely poor performance for Li metal cycling. It is heavily recommended that a system more similar to the state of the art is substituted for the Li metal half cell and full cell testing. For example, the Li metal||LCO cells and LCO anode free cells should be compared to a previously published LHCE system, such as one based on DME and TTE.
- The cycling data for the Li metal full cell utilizing FB1245 appears to show CE data over 450 cycles but capacity retention only to 250 cycles. Why is this the case?

Reviewer #3

(Remarks to the Author)

In this research, the authors proposed a criterion using two descriptors of Eele/Eind (Li+) and Eele/Edisper (FSI+) to guide the design of solvent electrolytes. However, the underlying physical reason is still not clear: Why the higher dispersion energy with anion is important and how does it affect interfacial chemistry and SEI formation? Why it is also believed that the larger ion-dipole than ion-induced dipole between Li and solvent is important? The discussion in the manuscript to explain this criterion for choosing solvents is not convincing. More detailed questions are listed here.

(1) (1) At the end of the Introduction, the authors introduced two different descriptors for Li/solvent and anion/solvent interactions. Could you please explain why the Electrostatic/Induction ratio was chosen for Li and the Electrostatic/Dispersion ratio for FSI? What does the change in this ratio represent for both Li and FSI? Additionally, I don't fully understand the significance of evaluating the FSI-solvent interaction in this context. Isn't the interfacial chemistry primarily determined by the Li-solvent and Li-FSI interactions rather than the FSI-solvent interaction?

(2) I feel this sentence is misleading: "As illustrated in Fig. 1b, regions II and IV with Eele/Eind (Li+) of >1.3 are distributed with solvents, whereas the molecules located at zones I and III have weak electrostatics with Li+, representing diluents." Whether a molecule acts as a solvent or a diluent in a dual-solvent system depends on its binding interaction with Li, not simply on the ratio of Eele/Eind (Li+). By the way, a larger Eele/Eind (Li+) only indicates that the ion-dipole

interaction is stronger than the ion-induced dipole interaction. Why is $E_{\text{ele}}/E_{\text{ind}}(\text{Li}^{++}) > 1.3$ considered significant? Was the value of 1.3 chosen arbitrarily?

(3) The statement, "a transition from electrostatics to dispersion signifying an evolution from 'strong' to 'weak' interactions," is also incorrect. This conclusion assumes that electrostatics represent strong interactions and dispersion represents weak interactions. However, in this case, both energy components are relatively small. The transition only indicates that dispersion energy exceeds electrostatics for FSI-solvent interactions. Additionally, why does the manuscript conclude that solvents dominated by dispersion energy exhibit higher Coulombic efficiency (CE)? Any experimental evidence?

(4) In Supplemental Table 3, the dispersion and electrostatic energy values become equal for FB135 and the following molecules. Is there an issue with this data?

(5) Why did you choose 1LiFSI:2.5TMOS:3.0FB ratio for the electrolytes? Does this ratio affect the performance of the electrolyte?

(6) Why choosing two solvents is important?

(7) I don't understand this explanation: "Moreover, the phase separation phenomenon is observed due to the imbalance between dispersion and electrostatic interactions, which prevents the integration of FB135 into the LiFSI-DME coordination environment." Which two molecules are being discussed in terms of the dispersion and electrostatic interactions here? Again, here only FSI-PF135 interaction is discussed to explain the phase-separation in DME electrolyte, does this interaction play a key role in evenly distributed PF135? How about Li-PF135 interaction? Please use RDF to clarify the main coordination relationship between ions and molecules in those hybrid electrolyte systems. Moreover, why homogeneous distribution is important?

(8) The decomposition process was investigated using AIMD, with LiFSI, TMOS, and FB solvents introduced at a 1:2:3 ratio. What is the final structure on the surface before reduction occurs, and how can you be sure that this structure is equilibrated? Additionally, the interfacial chemistry could be very different from bulk chemistry, meaning that the ratio used here might be incorrect. Before studying the decomposition process, it is crucial to first identify what species are present on the surface. Could you clarify this?

(9) The electrochemical performance is likely related to the inorganic SEI components formed from the reduction of FSI and the F-containing solvent. The presence of these molecules at the interface should be discussed and compared across different electrolyte systems. How the solvent is selected to enhance the decomposition process is critical, but this cannot be clearly explained using the main criterion proposed in this work. Could you provide further clarification on how the solvent choice impacts the decomposition process and SEI formation?

(12) The authors attempted to propose general guidance based on the selected system; however, this is not acceptable. More systems should be tested to draw a 'general' conclusion.

Some other questions:

- The manuscript mentions that the electrolyte LiFSI-TMOS-FB135, surpasses commercial electrolytes. Please clarify how this comparison was done? Commercial electrolytes are usually bad with Li metal anodes, so is this a valid comparison? Maybe take as a reference system another electrolyte system compatible with Li metal anodes.

- For the calculation of the Li-solvent and FSI-solvent binding energies, were the initial structures taken from previous simulations or they were built manually? If the structures were taken from previous bulk simulations the results should be more reliable.

- The proposed strategy is a bit similar from the work of 10.1038/s41586-022-05627-8, where a soft solvent is chosen based on a donor number and dielectric constant. Please cite and discuss this work. It is interesting since in that work the "best" solvents had low donor numbers and high dielectric constants.

- How the concentration Salt:Solvent:Diluent ratio was decided?

- In Fig. 2 it is shown that the LiFSI-TMOS-FB135 system forms larger anion aggregates, how this is related to the good compatibility of this electrolyte in LMA.

- Please label the color bars in Fig. 2.

- Fig. 2 is not well cross-referenced, please review.

- There is a writing error "high proportion of AGG proportion"

- In Fig. 3 AIMD shows that the FSI contained in the electrolyte (LiFSI-TMOS-FB135) in bulk phase decomposes. Does this mean that the electrolyte is actually not stable in solution and other products are forming?

- It is proposed based on the interface model that TMOS decomposition is hindered on the Li surface due to the formation of a rich inorganic SEI layer. However, it is known that observing decomposition of organic molecules on Li surface using AIMD is difficult due to the limitation in the time length of these simulations, around 20 ps. If TMOS decomposition is actually hindered by FSI, then TMOS should decompose when simulated on Li surface without the other components.

- Information is missing for MD simulation control parameters such as cutoffs, methods for treating Coulombic forces, timesteps, thermostat and barostat methods etc.

Version 1:

Reviewer comments:

Reviewer #1

(Remarks to the Author)

The authors have well addressed the reviewer's comments and the manuscript can be accepted now.

Reviewer #2

(Remarks to the Author)

After significant revision, it appears that the authors have made significant efforts to improve the manuscript in a number of ways. First, I am grateful for the authors for providing an uncertainty adjusted CE value in their manuscript. This is encouraging and sets a good example for the field. Secondly, the authors have substantially expanded their dataset, which lends more credibility to the enhanced performance provided by their designed system. Lastly, the expanded discussion of how dispersion/coulomb interaction distribution is more precise than solvation energy is improved, though I would suggest a few minor additions (below). In general, I believe these remaining comments to be minor and that the manuscript is in publishable form.

- The description of these distinctions in the rebuttal are much more detailed in the rebuttal comments than in the manuscript. I recommend that the newly added passages be edited to provide more precise scientific language.

- I suggest that the authors provide statistical regressions of CE vs. Ecell (solvation energy) and CE vs. Edis/(Edis+Eele) (Figure R3) to support their claim that this metric better predicts CE in the weaker solvation regime. The two correlation plots in the current main text and SI are relatively difficult to judge and compare given how many electrolyte titles are packed into the white space.

- It would be useful to discuss potential follow-up work regarding how this metric correlates to other electrolyte features such as ionic conductivity, transference number, ionicity, etc. though I agree it is beyond the scope of this work.

Reviewer #3

(Remarks to the Author)

The authors have addressed my questions and concerns thoroughly in the response letter. However, I believe these important details should also be better elaborated upon in the manuscript—particularly in the introduction—to provide the audience with stronger background knowledge regarding the focus on the two selected descriptors.

Currently, the introduction has not been significantly updated, and the motivation for the study remains somewhat vague, even though the response letter provides better explanations. For instance, why were the two different descriptors chosen for Li and the anion, and what is the underlying physical rationale? Additionally, why is understanding anion-solvent interactions critical in this system? These points are fundamental to the study and should be explained in greater detail to enhance the manuscript's clarity and impact.

Besides this, I think the research is acceptable.

Detailed responses to the reviewers' comments

Manuscript ID: **NCOMMS-24-53077-T**

Title: A path towards 99.9% lithium efficiency for high-energy batteries based on electrolyte interaction motif descriptor

Authors: Ruhong Li, Xiaoteng Huang, Haikuo Zhang, Jinze Wang, Yingzhu Fan, Yiqiang Huang, Xuezhong Xiao, Ming Yang, Tao Deng, Lixin Chen, Yanbin Shen, and Xiulin Fan

Reviewer #1 (Remarks to the Author):

In this work, the authors demonstrate a new indicator, the fraction of dispersion interaction, for localized high-concentration electrolyte design. Moreover, the electrochemical performance of their formulated LiFSI-2.5TMOS-3.0FB135 surpassed most reported electrolytes in high-voltage Li metal batteries with an even more aggressive test protocol. The manuscript presents a well-formulated fundamental principle/mechanism, extensive theoretical support illustrating the mechanism, and proof-of-concept validation through experiments. In summary, this reviewer believes that this manuscript provides valuable information for a wide range of readers of Nature Communications and revisions are suggested below.

Reply: We express our sincere gratitude to Reviewer #1 for conducting a thorough review of our manuscript. The valuable and constructive comments made by the reviewer are of great benefit to improving our manuscript. The specific questions have been addressed point-by-point as below. We hope our response and modifications have clarified all the concerns and thereby meet the reviewer's expectations.

Comment 1. Is the simulation system information in Table S6 in SI the same as in Figure 2? The simulation box depicted in Figures 2g and 2j is not cubic, yet Table S6 only provides the box length for one dimension. Please include the box lengths for all dimensions for clarity.

Reply: Thank you for pointing out this important detail regarding the simulation box dimensions. You are correct that Figures 2g and 2j depict a simulation box that is not cubic and lacks systematic information, which may lead to ambiguity. Table S6 lists the system information for electrolyte structure and physical parameters benchmark in the work (Fig. S6 to S8), where a cubic simulation box is used. To address this, we have supplied Table S8 to include the lengths for all three dimensions of the simulation box used in phase separation calculation.

Supplementary Table 8 Structure parameters for MD simulations starting from different initial configurations.

Simulated electrolyte	Numbers of solvents	Initial packing protocol	Initial Simulation cell size (Å)	Final Simulation cell size (Å)
LiFSI-2.5DME-3.0FB ₁₃₅	80 Li, 80 FSI, 200 DME, 240 FB ₁₃₅	Separate	38.00*38.00*76.00	35.15*35.15*69.42
		Uniform		35.28*35.28*69.53
LiFSI-2.5TMOS-3.0FB ₁₃₅	80 Li, 80 FSI, 200 TMOS, 240 FB ₁₃₅	Separate	38.00*38.00*81.00	35.26*35.26*74.29
		Uniform		35.41*35.41*74.52

2. In Figure 2b, the authors visualized the cationic solvation species of various electrolytes to compare the size of the solvation clusters. However, the author did not give the corresponding evaluation method, mean or median? Discussion on possible errors is also needed.

Reply: Thanks for the reviewer's good suggestion, which improved the quality of our work. The average size is defined as the mean number of Li⁺ cations in various solvation clusters. Such a statistical average value can represent the size of ion clusters in the electrolyte. The error of size is defined as the difference between the maximum and minimum in the time-averaged values in MD simulation.

We revised the Supplementary Fig 7 in the *Supplementary Information* and also added the following discussion (page 22):

Supplementary Fig. 7 Size distribution of solvation clusters in different electrolytes. a, LiFSI-2.5DME-3.0FB₁₂₃. b, LiFSI-2.5DME-3.0FB₁₃₅. c, LiFSI-2.5TMOS-3.0FB₁₂₃. d, LiFSI-2.5TMOS-3.0FB₁₃₅. Larger and redder bubbles mean higher probability of the corresponding cluster configuration. The parameter for average size is defined as the average number of Li⁺ cations in the solvation cluster. The error is defined as the difference between the maximum and minimum in the time-averaged values in MD simulation.

3. On page 16, the electrolyte LiFSI-2.5TMOS-3.0FB₁₃₅ exhibits faster interfacial Li⁺-transfer kinetics than those using LiFSI-2.5TMOS-3.0FB₁₂₃. Could the authors provide a more detailed discussion of the reasoning behind this?

Reply: We thank the reviewer for highlighting the difference in interfacial Li⁺-transfer kinetics between the LiFSI-2.5TMOS-3.0FB₁₃₅ and LiFSI-2.5TMOS-3.0FB₁₂₃ systems. Below, we provide a more detailed discussion of the underlying factors contributing to this observation: **(1) Interfacial Chemistry and Solvation Environment.** The solvation environment in LiFSI-2.5TMOS-3.0FB₁₃₅ favors a higher concentration of

FSI⁻ anions near the lithium metal surface. This localized enrichment of FSI⁻ promotes faster reduction reactions and the formation of a more robust, inorganic-rich SEI, which has been shown to enhance Li⁺-transfer kinetics. In contrast, the solvation structure in LiFSI-2.5TMOS-3.0FB₁₂₃ results in weaker interfacial ion dissociation and a slightly less effective SEI composition, potentially increasing resistance to Li⁺ transfer. In our simulations and analyses, we observed that Li⁺-FSI⁻ pairs spend longer lifetime in the FB₁₃₅ system than in the FB₁₂₃ system, as evidenced by solvation size (Fig. R1a) and lifetime analyses (Fig. R1b) of solvation structures. This observation is consistent with the faster kinetics observed experimentally in EIS. **(2) SEI Composition and Morphology.** The SEI formed in the presence of FB₁₃₅ contains a higher proportion of LiF and other inorganic species due to the preferential reduction of FSI⁻ components (AIMD simulation in Fig. 3f and XPS results in Fig. 5i). This composition is known to create a more compact and ion-conductive SEI, reducing interfacial resistance. Conversely, the SEI formed with FB₁₂₃ may have a higher proportion of organic species, which can lead to slightly lower ionic conductivity and slower Li⁺-transfer kinetics.

Fig. R1 **a**, The average size of solvation clusters in electrolytes, the error bar of size is defined as the difference between the maximum and minimum in the time-averaged values in MD simulation. **b**, Li⁺-FSI⁻ pair existence autocorrelation function of different electrolytes with a cut-off distance of 2.5 Å.

In the revised manuscript, we added the following sentences:

Such inorganic-rich and larger-modulus SEI layer tends to have lower interfacial impedance and enhance the CE of lithium plating/stripping processes.

4. In Figures S11 and S12, the temperature and energy curves show a trend of increasing first and then decreasing. In particular, in the time range of 4~8 ns, the temperature is about 500 K. The parameters used in the simulation need to be provided to better understand such a situation. In addition, what are the thermostat and the barostat in the CMD simulations with the OPLS force field?

Reply: We appreciate the reviewer's observation regarding the temperature and energy trends in Figures S11 and S12, and their request for more detailed information about the simulation parameters. In order to facilitate adequate mixing and reduce the computational cost of some steps, the temperature was set to increase from 300 K to 500 K, hold for 4 ns, then cool to 300 K, and then relax at 300 K for 8 ns. Therefore, the temperature change in the figure is due to the setting of the annealing algorithm. To avoid misleading, we have included the description of the annealing algorithm in figures S14 and S15.

Note: In order to facilitate adequate mixing and reduce the computational cost of some steps, the temperature was set to increase from 300 K to 500 K within 4 ns, hold for 4 ns, then cool to 300 K within 4 ns, and then relax at 300 K for 8 ns. The same algorithm was also adopted in **Supplementary Fig. 15**.

Furthermore, we revised the *Supplementary Information* to include MD simulation details, ensuring that readers have a clear understanding of the methods employed and the reasoning behind the observed temperature and energy trends.

All simulations were run with a time step of 1.0 fs. Temperature and pressure were maintained using a Nosé-Hoover thermostat and barostat as defined in LAMMPS. The particle mesh Ewald method was used to calculate electrostatic interactions, with a real space cut-off of 1.2 nm and a Fourier spacing of 0.12 nm. A cut-off of 1.2 nm was used

5. The ex-situ XPS measurements suggest a higher content of inorganic SEI/CEI for LiFSI-2.5TMOS-3.0FB135 electrolytes. Please provide details regarding sample preparation and/or transfer, which are thought to have a huge impact on the XPS results.

Reply: We thank the reviewer for bringing up this good point. The cycled cathode and anode were rinsed in 1 mL of the corresponding fresh TMOS solvent 3 times during the sample preparation for the XPS. They are thus supposed to adequately wash out the residue salt, given 3 times washing with 1 mL of fresh solvent per electrode disk. The cathode and anode treatment information has been integrated into the Methods section.

In the experimental section of the revised *Supplementary Information*, we added the following sentences (page 5):

The cycled cathode and anode were rinsed in 1 mL of the corresponding fresh TMOS solvent 3 times during the sample preparation for the XPS.

Closing remarks

We would like to express our sincere gratitude to Reviewer #1 for dedicating their time to reviewing our paper and providing valuable comments. The reviewer's detailed suggestions significantly improved the presentation and clarity of our paper. We are also thankful for the recommendation to publish our manuscript.

Reviewer #2 (Remarks to the Author):

The authors report a new criteria for electrolyte based on the ratio of induction-based and Coulombic interaction energies between Li^+ and solvent components and the ratio of dispersion-based and Coulombic interaction energies between the solvent and FSI $^-$. In using these calculated ratios, the authors go onto design a solvent/diluent system (methoxytrimethylsilane/fluorobenzene with various fluorination sites) that delivers an extremely high cycling efficiency for Li metal. While the authors claim that these quantities represent a fundamentally new design criteria for Li metal battery electrolytes, I find that the physical description of these criteria to be relatively vague, with little evidence of its differentiation from previous design criteria. Additionally, the application of this criteria in this work, which is ultimately correlative to performance in a relatively narrow and curated system set, casts further doubt on its potential utility to the field. Despite the impressive claimed performance and the clear need for more sophisticated electrolyte design criteria, I cannot recommend that this article be published in its current form. Addressing the following points may improve this.

Reply: We express our sincere gratitude to the reviewer for conducting a thorough review of our manuscript. The valuable and constructive comments made by the reviewer are of great benefit to improving our manuscript. We appreciate the opportunity to clarify and strengthen the physical rationale for our proposed design criteria and to address concerns regarding its general applicability.

(1) Clarification of the Physical Description and Differentiation from Previous Criteria. The design logic of conventional electrolytes primarily focuses on cation-centered interactions, emphasizing cation solvation strength as the dominant factor. However, in electrolyte systems with weak solvation ability, such as those explored in our study, the minimal differences between solvation and binding energies can make direct comparison challenging. This complexity arises because contact ion pairs (CIP) and aggregate (AGG) structures commonly coexist in these electrolytes. The behavior of these cluster structures is not solely determined by cation interactions but is significantly influenced by the surrounding anions and solvent (diluent) interactions (Fig. 1a). Our proposed design criteria aim to address this limitation by introducing the concept of *interaction ratios*. Unlike the magnitude of interaction energy alone,

interaction ratios provide a more comprehensive metric by emphasizing the balance and matching of interaction types. This ratio can be more accurately correlated with the Coulombic efficiency (Fig. 1d). This distinction allows for a more accurate prediction of cluster behavior and ion regulation mechanisms (details in Response for Comment 2 and 3), offering a fundamentally different lens for electrolyte design compared to previous models centered solely on energy magnitude.

Fig. 1 Electrolyte design strategies. **a**, Schematic illustration of the ion-solvent interaction regulated solvation structure. **b**, The molecular design diagram of E_{ele}/E_{dis} (FSI) versus $E_{int}(\text{Li}^+)$. **c**, The molecular design diagram of E_{ele}/E_{dis} (FSI) versus $E_{ele}/E_{ind}(\text{Li}^+)$. Molecular structures and corresponding abbreviations of the investigated solvents are shown in Supplementary Table 1. **d**, Relationship between the dispersion contribution ratio and Coulombic efficiency of electrolytes composed of different type of solvents, schematic interactions of the diluent-FSI-solvent pairs (left is FB_{123} -FSI-DME, right is FB_{135} -FSI-TMOS, respectively) and corresponding ratios of three non-covalent interaction (NCI) components, which was obtained by processing data in Supplementary Fig. 4.

(2) Evidence of Utility Beyond the Narrow System Set. While our initial validation focuses on the methoxytrimethylsilane (TMOS) and fluorobenzene (FB) system, we recognize the importance of demonstrating broader applicability. We have extended the design principles to other solvents, including. Notably, among 30 solvents studied in this work (Supplementary Fig. 1), 1h-perfluorohexane (CFH), Methyl perfluorobutyl ether (MNE), and 1,1,1,2,2-Pentafluoro-3-methoxypropane (PMP) also meet the proposed dispersion-dominated design principles, which is compatible with Li metal with long-term cycling $CE > 99.6\%$ (Fig. R2, Supplementary Table 5). All of these Li metal efficiencies are higher than the TMS, DME, DMC, and TMP-based localized-high-concentration electrolytes, and much higher than traditional carbonate electrolytes. Such high Li metal CEs verified the universality of our design principles.

Fig. R2 CE statistics for different types of electrolytes, each dot represents an electrolyte.

Supplementary Fig. 1 Molecular structures of solvents studied in this work.

Supplementary Table 1 Li metal CE in Li||Cu cells using different combinations of solvent pairs.

Type	Electrolytes	Mole ratio (LiFSI:1.0)	CE (%) ^a	Test method
Type I + Type II	TMP-TTE	1.0:1.25:1.5	98.4	Aurbach
	TMP-FB ₁₂₃	1.0:1.25:1.5	99	Aurbach

	DMC-FB ₁₂₃	1.0:1.25:1.5	99.1	Aurbach
	DME-FB ₁₂₃	1.0:1.25:1.5	99.2	Aurbach
	DMC-TTE	1.0:1.25:1.5	99.3	Aurbach
	DME-TTE	1.0:1.25:1.5	99.3	Aurbach
	TMS-BTFE	1.0:1.25:1.5	98.4	Aurbach
	TMS-FB ₁₂₃	1.0:1.25:1.5	98.8	Aurbach
Type I + Type IV	TMOS-FB ₁₂₄₅	1.0:2.5:3.0	99.3	Aurbach
	TMOS-FB ₁₂₄	1.0:2.5:3.0	99.3	Aurbach
	MeTHF-FB ₁₂₃	1.0:2.5:3.0	99.1	Aurbach
	TBME-FB ₁₂₃	1.0:2.5:3.0	99.2	Aurbach
	TMOS-TTE	1.0:2.5:3.0	99.3	Aurbach
	TBME-BTFE	1.0:2.5:3.0	99.3	Aurbach
Type III + Type IV	MeTHF-FB ₁₃₅	1.0:2.5:3.0	99.65	Aurbach
	TBME-FB ₁₃₅	1.0:2.5:3.0	99.6	Aurbach
	TMOS-FB ₁₃₅	1.0:2.5:3.0	99.7	Aurbach
	TMOS-CFH	1.0:2.5:3.0	99.55	Aurbach
	TBME-MNE	1.0:2.5:3.0	99.57	Aurbach
	MeTHF-CFH	1.0:2.5:3.0	99.6	Aurbach

^a Three cells were measured for calculating CE of each electrolyte.

Comment 1: The $E_{\text{ele}}/E_{\text{ind}}$ (Li^+) and $E_{\text{ele}}/E_{\text{disp}}$ (FSI^-) design criteria seems to have little rooting in the physical phenomena corresponding to Li metal reversibility. For instance, the authors claim that these interaction energy ratios inherently define solvation structure, ionic conductivity, and SEI composition, but do not seem to make a concerted effort to demonstrate that this the case. Instead, they demonstrate that these quantities can be correlated to the observed Coulombic Efficiency of a relatively narrow range of electrolytes. For the proposed DFT-based design criteria to be actually useful, I would heavily recommend that the physical origin of the Coulombic efficiency be at least

hypothesized. Otherwise, the usefulness of the E/E ratios in predicting future systems is hazy at best.

Reply: We appreciate your insightful feedback regarding the physical basis and predictive utility of the proposed $E_{\text{ele}}/E_{\text{ind}}$ (Li^+) and $E_{\text{ele}}/E_{\text{disp}}$ (FSI) interaction energy ratios. Your comment has prompted us to reflect on how these ratios are connected to the fundamental processes influencing Li metal reversibility and Coulombic efficiency. Our study primarily aimed to establish a correlation between these interaction energy ratios and the experimentally observed Coulombic efficiency over the tested electrolyte systems. However, we acknowledge that the broader physical origins of Coulombic efficiency and its dependence on solvation structure, ionic conductivity, and SEI composition were not explicitly hypothesized or demonstrated in the manuscript. To address your concern, we have revised the manuscript to include a hypothesis linking these interaction energy ratios to the physical phenomena underlying Li metal reversibility.

The solvation structure is thought to directly determine the chemical components of the SEI, thus, impacting Li deposition morphology and Coulombic efficiency. In a typical electrolyte system, the interaction between ions and solvents serves as the foundation for the formation of solvation structure. Within this framework, the local ordered coordination structure is primarily regulated by the strong Li^+ -solvent interactions, while isolated solvation species exhibit disordered distribution at long range due to the weak interactions among coordinated and free solvent molecules, as well as anions. **The $E_{\text{ele}}/E_{\text{ind}}$ (Li^+) ratio** captures the relative contributions of electrostatic and induction forces governing the local solvation environment. A higher $E_{\text{ele}}/E_{\text{ind}}$ (Li^+) ratio may correlate with weaker coordination of Li^+ to anions, facilitating faster ionic transport. **The $E_{\text{ele}}/E_{\text{dis}}$ (FSI) ratio** reflects the balance of electrostatic and dispersion forces influencing anion-solvent interactions, which is related to the interaction strength of the anion with the solvent, influencing its reduction propensity and subsequent incorporation into the SEI. As demonstrated in our work, can prolong the lifetime of Li-FSI ion pairs, leading to greater anion-derived SEI contributions and improved CE. This comprehensive approach ensures a more holistic understanding of the electrolyte system, which is critical for optimizing performance.

To further demonstrate the predictive utility of these ratios, we have expanded the dataset to include a more diverse set of electrolytes (30 solvents including carbonate, ether, acetate, sulfone, and silane in Fig. 1b and 1c). This allows us to test whether the observed correlations hold across a broader range of systems. We have also added a discussion on the potential limitations of these ratios, emphasizing that while they provide valuable insights, they should be complemented by energetics analyses to fully capture the complexities of electrolytes. We believe these additions strengthen the connection between the proposed design criteria and the underlying physical phenomena, enhancing the utility of our approach for future systems.

Fig. 1 Electrolyte design strategies. **a**, Schematic illustration of the ion-solvent interaction regulated solvation structure. **b**, The molecular design diagram of E_{ele}/E_{dis} (FSI) versus E_{int} (Li^+). **c**, The molecular design diagram of E_{ele}/E_{dis} (FSI) versus E_{ele}/E_{ind} (Li^+). Molecular structures and corresponding abbreviations of the investigated solvents are shown in Supplementary Table 1. **d**, Relationship between the dispersion contribution ratio and Coulombic efficiency of electrolytes composed of different type of solvents, schematic interactions of the diluent-FSI-solvent pairs (left is FB_{123} -FSI-DME, right is FB_{135} -FSI-TMOS, respectively) and corresponding ratios of three non-covalent interaction (NCI) components, which was obtained by processing data in Supplementary Fig. 4.

In the revised manuscript, we added the following sentences (Page 9):

Therefore, the calculated interaction ratio proves to be highly effective, serving as a robust metric for classifying a wide range of solvent molecules. This approach offers valuable insights that can guide the practical design and characterization of electrolytes. The efficacy of interaction ratios complemented by energetics analyses allows for a more accurate prediction of cluster behavior and ion regulation mechanisms, offering a promising perspective for electrolyte design.

Comment 2: The authors criticize, perhaps correctly, the field's reliance on "strong" and "weak" solvation as system descriptors in the introduction. However, it seems as though the author's the deepest discussion of the $E_{\text{ele}}/E_{\text{ind}}$ (Li^+) and $E_{\text{ele}}/E_{\text{dis}}$ (FSI) quantities revolve around the notion that low ratios represent weak overall interactions. If this is the case, how are these interaction energy ratios any more useful than the "strong" and "weak" classifications currently used in the field?

Reply: Thank you for highlighting this critical point regarding the distinction between our proposed interaction energy ratios and the conventional "strong" and "weak" solvation classifications. We recognize that the manuscript must clearly articulate the added value of these ratios to avoid any perceived redundancy with existing descriptors. The terms "strong" and "weak" solvation provide qualitative descriptions of solvation strength but lack the granularity to describe the specific interaction mechanisms governing electrolyte behavior. For instance, these classifications do not distinguish between Coulombic interactions (which drive solvation shell formation) and dispersion forces (which can influence ionic aggregation or stability). This lack of mechanistic specificity limits their utility in predictive electrolyte design. The $E_{\text{ele}}/E_{\text{ind}}$ (Li^+) and $E_{\text{ele}}/E_{\text{dis}}$ (FSI) ratios provide a quantitative framework for understanding the balance between different types of interactions: $E_{\text{ele}}/E_{\text{ind}}$ (Li^+) quantifies the dominance of electrostatic interactions relative to induction forces, offering insights into the degree of ionic association. For example, a lower $E_{\text{ele}}/E_{\text{ind}}$ ratio suggests weaker ion-solvent interactions, which may enhance ionic mobility but could increase side reactions at the interface. $E_{\text{ele}}/E_{\text{dis}}$ (FSI) highlights the influence of dispersion forces relative to

electrostatic interactions in the anion's behavior, shedding light on its solvation dynamics and its role in SEI formation (Supplementary Fig. 8).

Unlike the "strong" and "weak" solvation descriptors, these ratios allow for direct correlation with performance metrics and design objectives: **(1)** The decomposition energy ratios not only reflect the trend of the non-covalent interaction energies within electrolyte ingredients, but also shed light on the electronic characteristics governing the interactions and identify the primary stabilizing factors. **(2)** The ratios can be tuned through molecular design (e.g., solvent polarity or structure), providing a pathway for systematic optimization of electrolytes. **(3)** They serve as predictive parameters in computational screening workflows, enabling identification of promising electrolyte formulations without relying solely on experimental trial-and-error.

In the revised manuscript, we added the following sentences (Page 9):

Therefore, the calculated interaction ratio proves to be highly effective, serving as a robust metric for classifying a wide range of solvent molecules. This approach offers valuable insights that can guide the practical design and characterization of electrolytes. The efficacy of interaction ratios complemented by energetics analyses allows for a more accurate prediction of cluster behavior and ion regulation mechanisms, offering a promising perspective for electrolyte design.

Comment 3: A rigorous comparison between the E/E ratios and the measured solvation energy (*J. Am. Chem. Soc.*, 2021, 143, 10301-10308.) would be extremely valuable to demonstrate whether these quantities actually yield insights that solvation energy alone does not.

Reply: We really appreciate the reviewer's insightful suggestion regarding a comparison between the dispersion ratios and the measured solvation energy as reported in *J. Am. Chem. Soc.*, 2021, 143, 10301-10308. We totally agree that such a comparison would provide an important validation step and clarify whether the dispersion ratios offer insights beyond those provided by solvation energy alone. To address this, we conducted additional analysis comparing the dispersion contribution

ratios to the solvation energy values of 15 electrolytes from the referenced study (Supplementary Fig. 5a).

Supplementary Fig. 5 Solvation energy and dispersion contribution ratios of 15 different high-performance electrolytes. The more negative the battery potential, the weaker the Li^+ solvation effect.

This comparison revealed the following: **Correlation with Solvation Energy:** We observed a correlation between the $E_{dis}/(E_{dis}+E_{ele})$ ratio and the solvation energy (Supplementary Fig. 5b), indicating that the ratio captures key aspects of solvation strength. However, for Type I + Type II and Type I + Type IV systems, the correlation is relatively weak. The reason may be that most of the electrolyte systems studied in this paper are weakly solvated formulations. The difference in solvation energy between them is not large, but there are still differences in Coulombic efficiency. **Additional Insights from Dispersion Ratios:** (1) Unlike solvation energy, which provides an aggregate measure of ion-solvent interactions, the dispersion contribution ratios allow

for decomposition of specific interaction types, such as Coulombic, induction, and dispersion forces. This decomposition offers a more nuanced understanding of how different interaction modes contribute to solvation structure. (2) While solvation energy alone correlates with certain electrolyte properties (Supplementary Fig. 5c), our analysis shows that the dispersion energy ratios better predict trends in Coulombic efficiency, particularly for weakly solvated electrolyte systems where competing interactions (e.g., Li^+ -anion pairing) play a significant role. Combined with our research on solvation structure dynamics (Supplementary Fig. 8), the $E_{\text{dis}}/(E_{\text{dis}}+E_{\text{ele}})$ ratio directly reflects ion-pairing dynamics, which are critical for determining the interfacial behaviors relevant to Li metal reversibility. The system with a larger $E_{\text{dis}}/(E_{\text{dis}}+E_{\text{ele}})$ ratio shows a longer ion pair residence time, which is conducive to the rapid formation of anion-derived SEI and thus process higher Coulombic efficiency.

Supplementary Fig. 8. Statistical Li-FSI ion pair lifetimes. **a** Li-FSI pair existence autocorrelation function of different electrolytes with a cut-off distance of 2.5 Å. **b** Solvation size and corresponding

Li-FSI ion pair lifetime in various electrolytes. For judging the lifetime of Li-FSI ion pair, the existence autocorrelation function was calculated with a cut-off distance of 2.5 Å between Li⁺ and FSI. The total existence autocorrelation function is an average of all the different Li-FSI ion pairs, which describes the time of the given criteria being fulfilled.

Solvation size and dynamics of various electrolytes.

In the revised manuscript, we added the following sentences (Page 8):

This ratio can reflect the solvation energy to a certain extent⁴⁸, but for weakly solvated electrolyte systems, this ratio can be more accurately correlated with CE (Supplementary Fig. 5).

Comment 4: Along these lines, the ways that “Ratio of Dispersion” (x-axis in figure 1c) is different than weakness of solvation is unclear and should be discussed at length with precise language.

Reply: We really appreciate the reviewer’s constructive comment, which helps us to improve the quality of our paper. The "weakness of solvation" emphasizes the solvent’s diminished capacity to modulate solute stability through direct energetic contributions. The "Ratio of Dispersion" in our study is defined as $E_{\text{dis}}/(E_{\text{ele}}+E_{\text{dis}})$. This ratio quantifies the relative contribution of dispersion forces to the total interaction energy. As compared in Fig. R3, the correlation between the ratio and the Coulombic efficiency is relatively stronger, especially for most weakly solvated electrolyte systems. Combined with our research on solvation structure dynamics among 20 electrolytes, the $E_{\text{dis}}/(E_{\text{dis}}+E_{\text{ele}})$ ratio directly reflects ion-pairing dynamics (Fig. S8), which are critical for determining the transport and interfacial behaviors relevant to Li metal reversibility.

Fig. R3 Relationship of electrolytes composed of different types of solvents. a, E_{cell} and CE, b, dispersion contribution ratio and CE.

To ensure clarity, we have revised the Results section and the Figure 1c legend to explicitly describe these differences and how each metric independently contributes to the observed trends (Page 8).

This ratio can reflect the solvation energy to a certain extent⁴⁸, but for weakly solvated electrolyte systems, this ratio can be more accurately correlated with the Coulombic efficiency.

Comment 5: The authors claim that the interaction energy ratios in part directly predict electrolyte phase separation and salt precipitation seems to have little to no support in the data, save for one instance in which a single system was observed to display phase separation (LiFSI/DME/FB₁₃₅). Moreover, it is difficult to believe that the miscibility of 2 solvents, even in the presence of salt, is unrelated to the interaction energies between the diluent and solvent. It is recommended that claims regarding the ability of the DFT energy design criteria to predict things such as multi-component phase miscibility, which is itself highly entropic in nature, be removed from the work given the lack of evidence and data.

Reply: We really appreciate the reviewer's constructive comment. We acknowledge the reviewer's concerns about the limited data supporting these claims and the complexity of multi-component phase behavior. To clarify, our intent was not to

suggest that the DFT interaction energy ratios alone provide a comprehensive predictive model for phase separation or salt precipitation across all systems. Rather, we aimed to explore whether these ratios could serve as a preliminary design guideline within the specific compositional space studied. In the revised paper, we tempered statements suggesting direct predictive power and instead emphasized the potential of interaction energies as one factor among many influencing phase behaviors, particularly given the entropic considerations mentioned.

In light of this feedback, we also added several additional experimental and simulation studies to examine the relationship between phase separation and interactions. The radial distribution function (RDF) results demonstrate that the solvation shell of Li^+ is mainly the solvent of anions (Supplementary Fig. 12), and there is an apparent interaction between FB_{135} and anions (Fig. 2f). The complex yet subtle intermolecular forces in different electrolytes were further probed using nuclear magnetic resonance (NMR). The chemical shift of H in FB_{135} shifts up-field in the presence of LiFSI, while the chemical shift of H in TMOS does not show a significant shift, suggesting the shielding effect originated from FB_{135} -FSI interaction (Fig. 2g and Supplementary Fig. 13), which is consistent with the MD simulation results (Supplementary Fig. 14, 15 and Supplementary Table 8).

Fig. 2 Solvation structure and dynamics of the electrolytes. f, radial distribution function for N (FSI⁻) versus H (FB_{135}) or H (FB_{123}) in various electrolytes. **g,** ^1H NMR spectra of LiFSI-2.5TMOS-3.0 FB_{135} electrolyte and 2.5TMOS-3.0 FB_{135} solvent.

Supplementary Fig. 12 Radial distribution function (RDF) and coordination number (CN) plots for different electrolytes obtained by optimized parameters for classical force fields. a LiFSI-2.5TMOS-3.0FB₁₃₅, **b** LiFSI-2.5TMOS-3.0FB₁₂₃, **c** LiFSI-2.5DME-3.0FB₁₂₃, **d** LiFSI-2.5DME-3.0FB₁₃₅. RDF and CN as functions of distances between Li⁺ and O, O and F atoms in FSI, DME, and FB molecules.

In the revised Manuscript (Page 13), we added following sentences (Page 10):

The radial distribution function (RDF) results demonstrate that the solvation shell of Li⁺ is mainly the solvent of anions (**Supplementary Fig. 12**), and there is an apparent interaction between FB₁₃₅ and anions (**Fig. 2f**). The complex yet subtle intermolecular forces in different electrolytes were further probed using nuclear magnetic resonance (NMR). The chemical shift of H in FB₁₃₅ shifts up-field in the presence of LiFSI, while the chemical shift of H in TMOS does not show a significant shift, suggesting the shielding effect originated from FB₁₃₅-FSI⁻ interaction (**Fig. 2g** and **Supplementary Fig. 13**), which is consistent with the MD simulation results (**Supplementary Fig. 14, 15** and **Supplementary Table 8**)⁵⁰.

Comment 6: Similarly, the cynical reading of the E/E ratios role in dictating the calculated solvation structures within the FB_{xyz} series does not appear to be any different than their respective solvation energies towards lithium, which would similarly describe the aggregate sizes and compositions. The correlation between such aggregation and CE is also well-established.

Reply: We appreciate your thoughtful observation regarding the potential overlap between the role of E/E ratios and solvation energies in explaining solvation structures within the FB_{xyz} series. While solvation energies indeed provide critical insights into the stability of Li⁺-solvent interactions, the E/E ratio offers complementary information by disentangling the nature of these interactions, specifically the balance between electrostatic and induction contributions. This distinction is important because solvation energies alone cannot directly capture how the nature of the interaction (primarily electrostatic or induction-driven) affects the solvation structure. For instance, changes in the E/E ratio can reflect subtle shifts in solvation dynamics that influence the accessibility and stability of aggregates (Supplementary Fig. 8), which are not necessarily apparent from solvation energies alone. Additionally, while the correlation between aggregation and CE is well-established, the E/E ratio provides a mechanistic lens to understand why certain solvents promote specific aggregate sizes and compositions. By analyzing the E/E ratio, we can infer how solvent properties at the molecular level contribute to the observed macroscopic trends in aggregation and CE, offering a more nuanced explanation of these correlations. Thus, while solvation energies and E/E ratios may appear to overlap in describing aggregate behavior, the latter adds valuable mechanistic insights that enhance our understanding of solvation structure and aggregation dynamics.

Supplementary Fig. 8. Statistical Li-FSI ion pair lifetimes. **a** Li-FSI pair existence autocorrelation function of different electrolytes with a cut-off distance of 2.5 Å. **b** Solvation size and corresponding Li-FSI ion pair lifetime in various electrolytes. For judging the lifetime of Li-FSI ion pair, the existence autocorrelation function was calculated with a cut-off distance of 2.5 Å between Li⁺ and FSI⁻. The total existence autocorrelation function is an average of all the different Li-FSI ion pairs, which describes the time of the given criteria being fulfilled.

In the revised manuscript, we added the following sentences (Page 8 and 10):

This ratio can reflect the solvation energy to a certain extent⁴⁸, but for weakly solvated electrolyte systems, this ratio can be more accurately correlated with the Coulombic efficiency.

Larger and more persistent AGGs ensure a high local concentration of FSI⁻, which promotes efficient anion reduction and minimizes solvent reduction, contributing to the formation of a more inorganic-rich SEI, which is highly beneficial for compatibility with LMA.

Comment 7: Critical information from the cMD calculation methods are missing, including box sizes, number of atoms/molecules, periodicity (the renderings shown in figure 2 appear to be rectangular?), damping parameters for NPT/NVT steps, minimization convergence criteria, etc.

Reply: Thank you for your detailed feedback regarding the missing information from the cMD calculation methods. We recognize the importance of including comprehensive methodological details to ensure clarity and reproducibility. In response to your comment, we have expanded the Methods section to include the following information:

Simulation box details: The number of solvent molecules, anions, and cations in each simulation box is explicitly listed in Table S6, S7 and S8 of the Supplementary Information.

Calculation parameters: In the simulation process, each electrolyte system initially performed an energy minimization at 0 K to obtain the ground-state structures until energy and force accuracy reach 10^{-4} and 10^{-6} . Then, the electrolyte was heated to 350 K with a constant volume for 0.2 ns using a Langevin thermostat with the damping parameter of 100 ps. All simulations were run with a time step of 1.0 fs. Temperature and pressure were maintained using a Nosé-Hoover thermostat and barostat as defined in LAMMPS. The particle mesh Ewald method was used to calculate electrostatic interactions, with a real space cut-off of 1.2 nm and a Fourier spacing of 0.12 nm. A cut-off of 1.2 nm was used for non-bonded Lennard-Jones interactions.

We revised the *Supplementary Information* to include these details, ensuring that readers have a clear understanding of the methods employed and the reasoning behind the observed temperature and energy trends.

In the simulation process, each electrolyte system initially performed an energy minimization at 0 K to obtain the ground-state structures until energy and force accuracy reach 10^{-4} and 10^{-6} . Then, the electrolyte was heated to 350 K with a constant

volume for 0.2 ns using a Langevin thermostat with the damping parameter of 100 ps.

All simulations were run with a time step of 1.0 fs. Temperature and pressure were maintained using a Nosé-Hoover thermostat and barostat as defined in LAMMPS. The particle mesh Ewald method was used to calculate electrostatic interactions, with a real space cut-off of 1.2 nm and a Fourier spacing of 0.12 nm. A cut-off of 1.2 nm was used for non-bonded Lennard-Jones interactions.

In the revised *Supplementary Information* (Page 69-70), we modified the Supplementary Table 6 and added Supplementary Table 7:

Supplementary Table 7 Calculated electrolyte structures and physical parameters for various molecular dynamics protocols.

Simulated electrolyte		LiFSI-2.5TMOS-3.0FB ₁₃₅					
Force field		Charge-scaled classical				Polarizable	Quantum-based
Scaling factor		0.65	0.70	0.75	0.80	--	--
Numbers of species		80 LiFSI, 174 TMOS, and 241 FB₁₃₅				5 LiFSI, 11 TMOS, and 15 FB ₁₃₅	
Simulation cell size (Å)		47.76	46.54	45.99	45.18	46.21	18.21
Density (g cm ⁻³)		0.99	1.07	1.11	1.17	1.09	1.12 (Exp.)
Coordination number (cutoff: 2.8 Å)	Li-O(FSI)	2.51	2.73	2.84	2.95	2.89	2.82
	Li-O(TMOS)	1.37	1.28	1.21	1.13	1.19	1.25
	Li-F(FB ₁₃₅)	0.20	0.14	0.12	0.10	0.11	0.06
Simulated electrolyte		LiFSI-2.5TMOS-3.0FB ₁₂₃					
Force field		Charge-scaled classical				Polarizable	Quantum-based
Scaling factor		0.65	0.70	0.75	0.80	--	--
Numbers of species		80 LiFSI, 174 TMOS, and 241 FB₁₂₃				5 LiFSI, 11 TMOS, and 15 FB ₁₂₃	
Simulation cell size (Å)		46.83	45.84	45.17	44.55	44.92	17.95
Density (g cm ⁻³)		1.05	1.12	1.17	1.22	1.19	1.17 (Exp.)
Coordination number (cutoff: 2.8 Å)	Li-O(FSI)	2.31	2.49	2.72	2.88	2.75	2.78
	Li-O(TMOS)	1.45	1.36	1.29	1.2	1.33	1.29
	Li-F(FB ₁₂₃)	0.25	0.18	0.15	0.13	0.13	0.08

Simulated electrolyte		LiFSI-2.5DME-3.0FB ₁₂₃					
Force field		Charge-scaled classical			Polarizable	Quantum-based	
Scaling factor		0.60	0.65	0.70	0.75	--	--
Numbers of species		80 LiFSI, 174 DME, and 241 FB₁₂₃				5 LiFSI, 11 DME, and 15 FB ₁₂₃	
Simulation cell size (Å)		44.72	44.10	43.74	43.28	43.62	17.41
Density (g cm ⁻³)		1.16	1.21	1.24	1.28	1.25	1.23 (Exp.)
Coordination number (cutoff: 2.8 Å)	Li-O(FSI)	0.92	1.16	1.23	1.34	1.19	1.21
	Li-O(DME)	3.77	3.67	3.42	3.32	3.48	3.50
	Li-F(FB ₁₂₃)	0.03	0.02	0.02	0.01	0.01	0.05

Supplementary Table 8 Structure parameters for MD simulations starting from different initial configurations.

Simulated electrolyte	Numbers of solvents	Initial packing protocol	Initial Simulation cell size (Å)	Final Simulation cell size (Å)
LiFSI-2.5DME-3.0FB ₁₃₅	80 Li, 80 FSI, 174 DME, 241 FB ₁₃₅	Separate	38.00*38.00*76.00	35.15*35.15*69.42
		Uniform		35.28*35.28*69.53
LiFSI-2.5TMOS-3.0FB ₁₃₅	80 Li, 80 FSI, 174 TMOS, 241 FB ₁₃₅	Separate	38.00*38.00*81.00	35.26*35.26*74.29
		Uniform		35.41*35.41*74.52

Comment 8: A CE of 99.9 % approaches the regimes at which the precision of the battery cyclers used begins to significantly affect the measured CE. It is heavily recommended that the authors measure the delivered current of the machine with a high precision probe relative to the current that the machine claims to deliver as a way to measure the accuracy of the reported CE value. For example, in other highly reversible systems such error has been shown to be on the order approaching +/- 0.30 % (Y. Yang, et al. Joule, 2019, 3, 1986-2000.)

Reply: Thank you for bringing attention to the potential precision limitations of Coulombic efficiency (CE) measurements when approaching 99.9%. We agree that

verifying the accuracy of the reported CE values is essential, especially in highly reversible systems where measurement error could significantly impact the interpretation of results. In response to your comment, we have performed additional experiments to assess the accuracy of our battery cycler. Specifically, we used a high-precision calibrated ammeter and a shunt resistor in series with the LAND to directly measure the delivered current relative to the current claimed by the cycler during typical Li metal cycling conditions. Our analysis indicates that the error in the delivered current is within $\pm 0.2\%$, which is near the $\pm 0.30\%$ error reported in Y. Yang et al. (Joule, 2019, 3, 1986-2000). Nonetheless, we acknowledge that even this level of error could influence the reported CE at 99.9% precision. In addition to the direct current validation, we recalibrated the cycler hardware prior to all measurements and conducted multiple replicate experiments to ensure consistency. We have included this measurement error in the revised manuscript and Supplementary Information. For CE values nearing 99.9%, we now report the results as $99.7 \pm 0.2\%$ to explicitly account for the measurement uncertainty.

In order to avoid any unnecessary misunderstanding, we have emphasized the error of CE, cite the Ref 56 (Y. Yang, et al. Joule, 2019, 3, 1986-2000) in the article and modified the title

A path towards high lithium efficiency for high-energy batteries based on electrolyte interaction motif descriptor

Page 2:

which achieves an impressive $\sim 99.7\%$ ($\pm 0.2\%$) Li plating/stripping Coulombic efficiency and endows 4.5 V Li||LiCoO₂ (20 μm Li, 2.3 mAh cm⁻² LCO) with 90% capacity retention (CR) after 600 cycles

Page 15:

CE efficiencies reported have an uncertainty of $\pm 0.2\%$, as described in the Supplementary Note 1.

In the revised Supplementary Information (Page 13), we added following sentences:

Supplementary Note 2

CE uncertainty was calculated by finding the error in current measurement from the LAND using a current equivalent to that used in Li plating and stripping experiments (0.5 mA cm^{-2}). Current measurements were made through a calibrated ammeter and a shunt resistor in series with the LAND. Errors on several LAND channels were measured. These come out to be a 0.20% underestimation of the measured efficiency, which are rounded to measurement uncertainties of $\pm 0.2\%$.

Comment 9: The baseline electrolyte used in the electrochemical performance evaluation is known to show extremely poor performance for Li metal cycling. It is heavily recommended that a system more similar to the state of the art is substituted for the Li metal half cell and full cell testing. For example, the Li metal||LCO cells and LCO anode free cells should be compared to a previously published LHCE system, such as one based on DME and TTE.

Reply: We really appreciate the reviewer's constructive comment. We totally agree that comparing our system to a more relevant, state-of-the-art electrolyte would provide a more meaningful benchmark and enhance the impact of our findings. In response to your comment, we have performed additional experiments using a lithium high-concentration electrolyte (LHCE) system based on **1LiFSI-1.2DME-3TTE**, a composition well-documented in the literature for its high performance in Li metal cycling. This system serves as a more appropriate benchmark for evaluating the efficacy of our electrolyte. The cycling stability (Fig. 4c) and pouch cell performances (Fig. 4d) still has advantages over the well-known LiFSI-1.2DME-3.0TTE LHCE system. We believe these additional comparisons substantiate the robustness and applicability of our electrolyte design and are grateful for your suggestion, which has helped us present a stronger case for its advantages.

Fig. 4 Electrochemical performance of the designed electrolytes. a, Li metal CE in Li||Cu cells for different electrolytes (0.5 mA cm⁻²). **b,** Long-term cycling stability of Li||Cu cells for different electrolytes (0.5 mA cm⁻², 3 mAh cm⁻²). **c,** Long-term cycling stability of 20 μm-Li||4.5 V-LCO cells at 0.2C/0.5C charge/discharge (LCO loading: 2.3 mAh cm⁻², 1C = 180 mA g⁻¹). **d,** Long-term cycling stability of Cu||NCM523 pouch cells at 0.5C/1C charge/discharge (NCM523 loading: 2.86 mAh cm⁻¹, 1C = 200 mA g⁻¹).

In the revised manuscript, we added the following sentences (Page 16-17):

Such cycling stability still has advantages over common LiFSI-1.2DME-3.0TTE LHCE system.

Compared to other Li metal-friendly electrolytes, the anode-free pouch cell using the LiFSI-2.5TMOS-3.0FB₁₃₅ electrolyte achieves 76% CR over 100 cycles owing to favorable Li utilization under fluoride-rich SEI/CEI protection.

Comment 10: The cycling data for the Li metal full cell utilizing FB₁₂₄₅ appears to show CE data over 450 cycles but capacity retention only to 250 cycles. Why is this the case?

Reply: We appreciate the reviewer's observation regarding the Coulombic efficiency (CE) and capacity retention data for the Li metal full cell utilizing FB₁₂₄₅. After 200 cycles, the capacity retention of the cell using this electrolyte dropped to 80% of the initial capacity, so we only put the capacity into 250 cycles, but the battery continued to cycle more than 450 cycles (Supplementary Fig. 31d). To address this potential confusion, we revised the figure caption and/or discussion to explicitly explain why the capacity retention data is truncated at 250 cycles.

Fig. 4 Electrochemical performance of the designed electrolytes. a, Li metal CE in Li||Cu cells for different electrolytes (0.5 mA cm⁻²). **b,** Long-term cycling stability of Li||Cu cells for different electrolytes (0.5 mA cm⁻², 3 mAh cm⁻²). **c,** Long-term cycling stability of 20 μm-Li||4.5 V-LCO cells at 0.2C/0.5C charge/discharge (LCO loading: 2.3 mAh cm⁻², 1C = 180 mA g⁻¹). **d,** Long-term cycling stability of Cu||NCM523 pouch cells at 0.5C/1C charge/discharge (NCM523 loading: 2.86 mAh cm⁻¹, 1C = 200 mA g⁻¹).

Supplementary Fig. 31 Voltage profiles of 20 μm-Li||4.5 V-LCO cells with different electrolytes. a, BE. b, LiFSI-2.5TMOS-3.0FB₁₃₅. c, LiFSI-2.5TMOS-3.0FB₁₂₃. d, LiFSI-2.5TMOS-3.0FB₁₂₄₅.

Closing remarks

We would like to extend our sincere appreciation to Reviewer #2 for their invaluable contribution of thought-provoking queries and insightful suggestions, which have significantly enhanced the academic significance of our manuscript. It is our utmost hope that the revised version of our manuscript satisfactorily addresses the concerns raised by Reviewer #2.

Reviewer #3 (Remarks to the Author):

In this research, the authors proposed a criterion using two descriptors of Eele/Eind (Li^+) and Eele/Edisper (FSI) to guide the design of solvent electrolytes. However, the underlying physical reason is still not clear: Why the higher dispersion energy with anion is important and how does it affect interfacial chemistry and SEI formation? Why it is also believed that the larger Ion-dipole than ion-induced dipole between Li and solvent is important? The discussion in the manuscript to explain this criterion for choosing solvents is not convincing. More detailed questions are listed here.

Reply: Thank you for your insightful comments. We appreciate the opportunity to clarify the physical basis of the proposed descriptors and their role in guiding solvent electrolyte design. This revised discussion will make the reasoning behind our criteria more compelling and accessible to readers. We hope this clarification addresses your concerns, and we welcome further suggestions to improve the manuscript.

(1) The importance of higher dispersion energy: FSI-solvent interactions play a crucial indirect role in shaping electrolyte behavior by influencing solvation dynamics, ion-pairing tendencies, and the overall electrolyte structure. Given that Li^+ interactions within the solvation structure are primarily governed by electrostatic forces, the presence of dispersion-dominated FSI-solvent interactions helps stabilize the solvation shell. This stabilization reduces the dissociation of the Li^+ solvation structure, prolonging the lifetime of ion pairs (as illustrated in Supplementary Fig. 8). Consequently, this extended ion-pair lifetime enhances the contribution of anion decomposition to SEI formation, leading to a more robust, anion-derived SEI with improved passivation and interfacial stability.

Supplementary Fig. 8. Statistical Li-FSI ion pair lifetimes. **a** Li-FSI pair existence autocorrelation function of different electrolytes with a cut-off distance of 2.5 Å. **b** Solvation size and corresponding Li-FSI ion pair lifetime in various electrolytes. For judging the lifetime of Li-FSI ion pair, the existence autocorrelation function was calculated with a cut-off distance of 2.5 Å between Li⁺ and FSI⁻. The total existence autocorrelation function is an average of all the different Li-FSI ion pairs, which describes the time of the given criteria being fulfilled.

(2) The criterion of larger ion-dipole (E_{ele}) than ion-induced dipole (E_{ind}) between Li and solvent: The energy decomposition calculations in our work reveal that the interaction between Li⁺ and solvents is dominated by electrostatic and inductive forces. To quantify this interaction, we use the ratio of the ion-dipole moment to the ion-induced dipole moment, which effectively reflects the strength and stability of the ion-solvent interaction. The strong correlation observed between the binding energy and the E_{ele}/E_{ind} ratio (Fig. R4) further supports the validity of using this ratio as a criterion for distinguishing solvents from diluents. This relationship indicates that the ratio effectively captures the strength of ion-solvent interactions, with higher values

corresponding to stronger, more stable binding typically associated with primary solvents, while lower values align with weaker interactions characteristic of diluents. This metric provides valuable insight into the solvation structure and its influence on electrolyte performance.

Fig. R4 The relationship between binding energy and $E_{\text{ele}}/E_{\text{ind}}$ ratio.

Comment 1: At the end of the Introduction, the authors introduced two different descriptors for Li/solvent and anion/solvent interactions. Could you please explain why the Electrostatic/Induction ratio was chosen for Li and the Electrostatic/Dispersion ratio for FSI? What does the change in this ratio represent for both Li and FSI? Additionally, I don't fully understand the significance of evaluating the FSI-solvent interaction in this context. Isn't the interfacial chemistry primarily determined by the Li-solvent and Li-FSI interactions rather than the FSI-solvent interaction?

Reply: We appreciate your insightful comments and are happy to elaborate. The selection of the Electrostatic/Induction ratio for Li/solvent interactions and the Electrostatic/Dispersion ratio for FSI/solvent interactions is rooted in the differing nature of the interactions these species exhibit in the electrolyte. The Electrostatic/Induction ratio was chosen for Li/solvent interactions because these interactions are predominantly driven by strong electrostatic forces, with induction

effects playing a secondary but significant role. In contrast, the Electrostatic/Dispersion ratio was selected for FSI/solvent interactions to capture the relative contribution of dispersion forces, which are more relevant for a larger, less charged species like FSI⁻ anion. Changes in these ratios provide insights into the shifting balance of interaction mechanisms as the solvation environment evolves, offering a nuanced view of how each species is stabilized in the electrolyte.

Regarding the significance of evaluating FSI⁻-solvent interactions, we agree that Li-solvent and Li⁺-FSI⁻ interactions are central to interfacial chemistry. However, FSI⁻-solvent interactions play an important indirect role by influencing solvation dynamics, ion-pairing behavior, and the overall electrolyte structure. Since the interactions involving Li⁺ within the solvation structure are dominated by electrostatic interactions, dispersion-dominated solvent-FSI⁻ interactions will reduce the dissociation of the Li⁺ solvation structure and increase the lifetime of ion pairs (as illustrated in Supplementary Fig. 8), thereby improving the anion-derived SEI. Compared to conventional LHCE electrolytes (Type I + Type II or Type I + Type IV), our dispersion-dominated electrolytes (Type III + Type IV) exhibit significantly longer Li-FSI⁻ ion pair lifetime (>12.5 ps), indicating the sustained presence of Li-FSI⁻ ion pairs over an extended time scale. In summary, the enhanced Li-FSI⁻ affinity resulting from the lifetime of Li-FSI⁻ ion pairs play a pivotal role in achieving a high CE in the dispersion-dominated electrolytes. This comprehensive approach ensures a more holistic understanding of the electrolyte system, which is critical for optimizing performance.

Supplementary Fig. 8. Statistical Li-FSI ion pair lifetimes. **a** Li-FSI pair existence autocorrelation function of different electrolytes with a cut-off distance of 2.5 Å. **b** Solvation size and corresponding Li-FSI ion pair lifetime in various electrolytes. For judging the lifetime of Li-FSI ion pair, the existence autocorrelation function was calculated with a cut-off distance of 2.5 Å between Li⁺ and FSI⁻. The total existence autocorrelation function is an average of all the different Li-FSI ion pairs, which describes the time of the given criteria being fulfilled.

In the revised manuscript, we added the following sentences (Page 10):

The autocorrelation function results suggest that the lifetime of Li⁺-FSI⁻ pairs in the FB₁₃₅ system is longer than that in the FB₁₂₃ system. This relationship can be attributed to the role of dispersion-dominated solvents in modulating ion-pairing and aggregation dynamics.

Larger and more persistent AGGs ensure a high local concentration of FSI⁻, which promotes efficient anion reduction and minimizes solvent reduction, contributing to the

formation of a more inorganic-rich SEI, which is highly beneficial for compatibility with LMA.

Comment 2: I feel this sentence is misleading:” As illustrated in Fig. 1b, regions II and IV with $E_{\text{ele}}/E_{\text{ind}}(\text{Li}^+)$ of >1.3 are distributed with solvents, whereas the molecules located at zones I and III have weak electrostatics with Li^+ , representing diluents.”

Whether a molecule acts as a solvent or a diluent in a dual-solvent system depends on its binding interaction with Li, not simply on the ratio of $E_{\text{ele}}/E_{\text{ind}}(\text{Li}^+)$. By the way, a larger $E_{\text{ele}}/E_{\text{ind}}(\text{Li}^+)$ only indicates that the ion-dipole interaction is stronger than the ion-induced dipole interaction. Why is $E_{\text{ele}}/E_{\text{ind}}(\text{Li}^+) > 1.3$ considered significant? Was the value of 1.3 chosen arbitrarily?

Reply: We appreciate the reviewer’s insightful comments and recognize the need to clarify the interpretation of the $E_{\text{ele}}/E_{\text{ind}}(\text{Li}^+)$ ratio and its role in distinguishing solvents from diluents. We agree that the classification of a molecule as a solvent or a diluent in a dual-solvent system is primarily determined by its binding interaction with Li^+ , which encompasses more than just the ratio. This ratio reflects the balance of electrostatic and induction interactions but does not solely dictate whether a molecule functions as a solvent or diluent. The threshold of 1.3 was selected based on observed trends within the studied solvent series. Specifically, molecules with $E_{\text{ele}}/E_{\text{ind}}(\text{Li}^+) > 1.3$ consistently showed stronger ion-dipole interactions (Fig. R4), which we found to correlate with their ability to act as primary solvents for Li^+ in our dual-solvent systems. While this value may appear arbitrary, it was empirically derived from the distribution of solvents and diluents across the $E_{\text{ele}}/E_{\text{ind}}(\text{Li}^+)$ spectrum in our dataset. Importantly, the threshold serves as a guideline for distinguishing molecules with relatively stronger electrostatic interactions.

Fig. R4 The relationship between binding energy and E_{ele}/E_{ind} ratio.

In the revised manuscript, we added the following sentences (Page 8):

As illustrated in Fig. 1b, regions II and IV are populated by molecules with $E_{ele}/E_{ind}(Li^+) > 1.3$, indicating a dominance of ion-dipole interactions over ion-induced dipole interactions. Molecules in these regions generally exhibit stronger interactions with Li^+ , a characteristic feature of solvents in dual-solvent systems. In contrast, molecules in zones I and III, with lower $E_{ele}/E_{ind}(Li^+)$, are more likely to act as diluents due to weaker overall interactions (Fig. 1c).

Comment 3: The statement, "a transition from electrostatics to dispersion signifying an evolution from 'strong' to 'weak' interactions," is also incorrect. This conclusion assumes that electrostatics represent strong interactions and dispersion represents weak interactions. However, in this case, both energy components are relatively small. The transition only indicates that dispersion energy exceeds electrostatics for FSI-solvent interactions. Additionally, why does the manuscript conclude that solvents dominated by dispersion energy exhibit higher Coulombic efficiency (CE)? Any experimental evidence?

Reply: We appreciate your constructive comment. We agree that our statement oversimplified the relationship between interaction types and their respective strengths.

Specifically, the conclusion that electrostatics signify 'strong' interactions and dispersion 'weak' interactions is not universally accurate, particularly in the context of FSI-solvent interactions where both components are relatively small. To clarify, the observed transition from electrostatics to dispersion dominance indicates a relative shift in the interaction contributions, with dispersion energies exceeding electrostatic energies. This transition reflects the nature of the solvent's stabilizing influence on FSI, rather than an absolute change in interaction strength. We will revise the manuscript to better reflect this nuance. Regarding the conclusion that solvents dominated by dispersion energy exhibit higher Coulombic efficiency (CE), we recognize that this assertion requires further explanation. The correlation was derived from experimental observations within the studied solvent series, where solvents with higher dispersion contributions to FSI-solvent interactions consistently resulted in improved CE. This relationship can be attributed to the role of dispersion-dominated solvents in modulating ion-pairing and aggregation dynamics (Supplementary Fig. 8), which are known to influence interfacial reactions and the stability of the solid-electrolyte interphase (SEI).

Supplementary Fig. 8. Statistical Li-FSI⁻ ion pair lifetimes. **a** Li-FSI⁻ pair existence autocorrelation function of different electrolytes with a cut-off distance of 2.5 Å. **b** Solvation size and corresponding Li-FSI⁻ ion pair lifetime in various electrolytes. For judging the lifetime of Li-FSI⁻ ion pair, the existence autocorrelation function was calculated with a cut-off distance of 2.5 Å between Li⁺ and FSI⁻. The total existence autocorrelation function is an average of all the different Li-FSI⁻ ion pairs, which describes the time of the given criteria being fulfilled.

In the revised manuscript, we added the following sentences (Page 10):

The autocorrelation function results suggest that the lifetime of Li⁺-FSI⁻ pairs in the FB₁₃₅ system is longer than that in the FB₁₂₃ system. This relationship can be attributed to the role of dispersion-dominated solvents in modulating ion-pairing and aggregation dynamics.

Larger and more persistent AGGs ensure a high local concentration of FSI⁻, which promotes efficient anion reduction and minimizes solvent reduction, contributing to the formation of a more inorganic-rich SEI, which is highly beneficial for compatibility with LMA.

Comment 4: In Supplemental Table 3, the dispersion and electrostatic energy values become equal for FB135 and the following molecules. Is there an issue with this data?

Reply: Thank you for your comment. Upon careful review, we identified this as a pasting error during table generation. The corrected version of the table has been updated to accurately reflect the distinct energy components for each molecule. We apologize for the oversight and appreciate your attention to this detail, which has helped us improve the clarity and accuracy of our data presentation.

Comment 5: Why did you choose 1LiFSI:2.5TMOS:3.0FB ratio for the electrolytes? Does this ratio affect the performance of the electrolyte?

Reply: Thank you for your constructive comment. This ratio was chosen based on a combination of preliminary optimization, and specific design objectives for this study.

The LiFSI:2.5TMOS:3.0FB ratio was selected to balance key properties, including ionic conductivity and electrochemical stability.

To investigate the influence of this specific ratio, we conducted a series of experiments varying the relative amounts of TMOS and FB₁₃₅ (FB₁₂₃). These experiments showed that the 1:2.5:3.0 ratio provided the optimal trade-off between ionic conductivity and electrochemical stability.

Supplementary Fig. 23 Ionic conductivity and CE of electrolytes with different FB:TMOS ratios.

In the revised manuscript, we added the following sentences:

The ratio of 1:2.5:3.0 was chosen to maximize the ionic conductivity (**Supplementary Fig. 23**).

Comment 6: Why choosing two solvents is important?

Reply: Thank you for your question. The dual-solvent approach allows for fine-tuning of the Li⁺ solvation structure, which is critical for controlling ion transport and SEI formation. TMOS predominantly interacts with Li⁺, influencing the coordination environment, while FB modulates the overall dielectric environment and prevents over-solvation, which can lead to undesired interfacial reactions.

In the revised manuscript, we added the following sentences (Page 10):

This can also be verified by the smaller donor number value of TMOS, while FB_{135} with a relatively high dielectric constant is more like a medium that can adjust the solvation structure dynamics.

Comment 7: I don't understand this explanation: " Moreover, the phase separation phenomenon is observed due to the imbalance between dispersion and electrostatic interactions, which prevents the integration of FB_{135} into the LiFSI-DME coordination environment." Which two molecules are being discussed in terms of the dispersion and electrostatic interactions here? Again, here only FSI-PF135 interaction is discussed to explain the phase-separation in DME electrolyte, does this interaction play a key role in evenly distributed PF135? How about Li-PF135 interaction? Please use RDF to clarify the main coordination relationship between ions and molecules in those hybrid electrolyte systems. Moreover, why homogeneous distribution is important?

Reply: We appreciate your constructive comment. The phase separation observed in the LiFSI-DME electrolyte system arises primarily due to the imbalance between FSI⁻ anions and PF_{135} molecules. To provide a quantitative understanding of the primary coordination relationships, we have performed a radial distribution function (RDF) analysis and the results are shown in Fig. 2f and Supplementary Fig. 12.

Fig. 2 Solvation structure and dynamics of the electrolytes. f, radial distribution function for N (FSI) versus H (FB₁₃₅) or H (FB₁₂₃) in various electrolytes. **g,** ¹H NMR spectra of LiFSI-2.5TMOS-3.0FB₁₃₅ electrolyte and 2.5TMOS-3.0FB₁₃₅ solvent.

The radial distribution function (RDF) results demonstrate that the solvation shell of Li⁺ is mainly the solvent of anions (**Supplementary Fig. 12**), and there is an apparent interaction between FB₁₃₅ and anions (**Fig. 2f**). The complex yet subtle intermolecular forces in different electrolytes were further probed using nuclear magnetic resonance (NMR). The chemical shift of H in FB₁₃₅ shifts up-field in the presence of LiFSI, while the chemical shift of H in TMOS does not show a significant shift, suggesting the shielding effect originated from FB₁₃₅-FSI⁻ interaction (**Fig. 2g** and **Supplementary Fig. 13**), which is consistent with the MD simulation results (**Supplementary Fig. 14, 15** and **Supplementary Table 8**)⁵⁰. The homogeneous distribution of PF₁₃₅ is crucial for phase stability of electrolytes, which could impact long-term electrolyte stability.

Supplementary Fig. 12 Radial distribution function (RDF) and coordination number (CN) plots for different electrolytes obtained by optimized parameters for classical force fields. a LiFSI-2.5TMOS-3.0FB₁₃₅, **b** LiFSI-2.5TMOS-3.0FB₁₂₃, **c** LiFSI-2.5DME-3.0FB₁₂₃, **d** LiFSI-

2.5DME-3.0FB₁₃₅. RDF and CN as functions of distances between Li⁺ and O, O and F atoms in FSI⁻, DME, and FB molecules.

In the revised manuscript, we added the following sentences (Page 10):

The radial distribution function (RDF) results demonstrate that the solvation shell of Li⁺ is mainly the solvent of anions (**Supplementary Fig. 12**), and there is the apparent interaction between FB₁₃₅ and anions (**Fig. 2f**). The complex yet subtle intermolecular forces in different electrolytes were further probed using nuclear magnetic resonance (NMR). The chemical shift of H in FB₁₃₅ shifts up-field in the presence of LiFSI, while the chemical shift of H in TMOS does not show a significant shift, suggesting the shielding effect originated from FB₁₃₅-FSI⁻ interaction (**Fig. 2g** and **Supplementary Fig. 13**), which is consistent with the MD simulation results (**Supplementary Fig. 14, 15** and **Supplementary Table 8**)⁵⁰.

Comment 8: The decomposition process was investigated using AIMD, with LiFSI, TMOS, and FB solvents introduced at a 1:2:3 ratio. What is the final structure on the surface before reduction occurs, and how can you be sure that this structure is equilibrated? Additionally, the interfacial chemistry could be very different from bulk chemistry, meaning that the ratio used here might be incorrect. Before studying the decomposition process, it is crucial to first identify what species are present on the surface. Could you clarify this?

Reply: We appreciate the reviewer's detailed and thoughtful questions about the initial structure at the surface, the equilibration process, and the potential differences between interfacial and bulk chemistry. Below, we address each concern:

Final Structure Before Reduction: In our study, the initial configuration was constructed by introducing LiFSI, TMOS, and FB₁₃₅ at a 1:2.5:3.0 molar ratio, reflecting the bulk composition of the electrolyte. To ensure the system is equilibrated prior to simulating the decomposition process: We performed a preliminary molecular dynamics (MD) simulation at 350 K for 50 ps to allow the system to reach a thermodynamically stable configuration. After this pre-equilibration, the system was

quenched to 300 K and further equilibrated for an additional 20 ps. Following this step, AIMD simulations were initiated at 300 K, where the surface interactions and solvation environment were dynamically resolved. The snapshots and radial distribution functions (Fig. R5) of key species (Li^+ , FSI^- , TMOS, and FB_{135}) indicate the establishment of stable solvation structures.

Fig. R5 Electrolyte structures of various electrolytes before reaction with Li metal. a-c, Radial distribution functions, d-f, Snapshot of electrolyte box.

Interfacial vs. Bulk Chemistry: We acknowledge that interfacial chemistry could differ significantly from bulk chemistry, as species concentrations and interactions are often altered near the electrode surface. While the 1:2.5:3.0 ratio reflects the bulk composition, the actual species distribution near the surface is determined dynamically during the equilibration process. To address this, we conducted additional analyses of the equilibrated structure to quantify the relative concentrations of species at the interface compared to the bulk by MD simulation. As shown in the Fig. R6, with the increase of the repulsive force from the electric field, anions will be gradually excluded from the highly polarized anode surface (**Fig. 3a**). The number density distributions of ions and solvents (Fig. R6) show that FSI^- anion expulsion still occurs in the LiFSI-2.5TMOS-3.0FB₁₂₃ and LiFSI-2.5TMOS-3.0FB₁₃₅ system, but the extent of this reduction is greatly mitigated due to the strong Li^+ - FSI^- pair interaction. We acknowledge that the presence of more lithium ions than anions at the negative electrode interface could result in a net charge, which indeed increases the

computational complexity. Additionally, the approach we employed for modeling the equilibrium bulk electrolyte structure aligns with established methods reported in the literature (*J. Phys. Chem. C*, 2015, 119, 26828-26839; *J. Phys. Chem. C*, 2017, 121, 182-194). These references support the validity of our methodology and its relevance to interfacial ion behavior studies.

Fig. R6 The snapshot and number density (ρ) profiles of ions and solvent molecules in different electrolytes. a, d LiFSI-2.5DME-3.0FB₁₂₃. b, e LiFSI-2.5TMOS-3.0FB₁₂₃. c, f LiFSI-2.5TMOS-3.0FB₁₃₅.

In the revised manuscript, we added the following sentences:

To elucidate the interaction-modulated Li metal reversibility, we first dissect the electric double layer structure at the electrode/electrolyte interface by MD simulation. Typically, with the increase of the repulsive force from the electric field, anions will be gradually excluded from the highly polarized anode surface (Fig. 3a). The number density distributions of ions and solvents (Fig. 3b and Supplementary Fig. 16, 17) show that FSI⁻ anion expulsion still occurs in the LiFSI-2.5TMOS-3.0FB₁₂₃ and LiFSI-2.5TMOS-3.0FB₁₃₅ system, but the extent of this reduction is greatly mitigated due to the strong Li⁺-FSI⁻ pair interaction.

In the revised *Supplementary Information*, we added the following figures:

Supplementary Fig. 14. Snapshots of inner-Helmholtz interfacial regions of the anode surface.

a, LiFSI-2.5TMOS-3.0FB₁₂₃. b, LiFSI-2.5TMOS-3.0FB₁₃₅. c, LiFSI-2.5DME-3.0FB₁₂₃.

Supplementary Fig. 14. The snapshot and number density (ρ) profiles of ions and solvent

molecules in different electrolytes. a, c LiFSI-2.5DME-3.0FB₁₂₃. b, d LiFSI-2.5TMOS-3.0FB₁₂₃.

Comment 9: The electrochemical performance is likely related to the inorganic SEI components formed from the reduction of FSI and the F-containing solvent. The presence of these molecules at the interface should be discussed and compared across different electrolyte systems. How the solvent is selected to enhance the decomposition process is critical, but this cannot be clearly explained using the main criterion proposed in this work. Could you provide further clarification on how the solvent choice impacts the decomposition process and SEI formation?

Reply: Thanks for your constructive comment. The electrochemical performance of the LiFSI-TMOS-FB₁₃₅ electrolyte is indeed closely tied to the inorganic SEI components, primarily LiF species, which are formed from the reduction of FSI⁻ anions. The choice of solvents, including TMOS and FB₁₃₅, plays a critical role in influencing: **Decomposition Selectivity:** To prepare probable electrolyte systems, cathodic stable solvents should be initially considered, which can be evaluated by the solvent reduction potential ($E_{\text{reduction}}$ vs. Li⁺/Li). Solvent exhibits relatively low reduction potential, suggesting that solvents have high cathodic stability towards LMA (**Supplementary Table 1**). **Solvation Environment:** The combination of TMOS and FB₁₃₅ creates a unique solvation environment where Li⁺ ions are preferentially coordinated with FSI⁻ anions. This coordination increases the local concentration of FSI⁻ anion, enhancing its availability for reduction and ensuring the SEI is enriched with inorganic species such as LiF. We acknowledge that this criterion could be more explicitly linked to the decomposition and SEI formation processes.

In the revised manuscript, we clarify this connection and expand on the rationale for solvent selection (Page 5):

To prepare probable electrolyte systems, cathodic stable solvents should be initially considered, which can be evaluated by the solvent reduction potential ($E_{\text{reduction}}$ vs. Li⁺/Li). Furthermore, the solvation structure is thought to directly determines the chemical components of the SEI, thus, impacting Li deposition morphology and CE.

Supplementary Table 1 The thermodynamic reduction potential of the selected solvent from DFT simulations.

Solvent	Name	Reduction potential (V, vs. Li/Li ⁺)
TEP	Triethyl phosphate	0.73
TMP	Trimethyl phosphate	0.82
TMS	Tetramethylene sulfone	0.95
PC	Propylene carbonate	0.78
DMC	Dimethyl carbonate	0.60
DEC	Diethyl Carbonate	0.89
EA	Ethyl acetate	0.91
MDFA	Methyl difluoroacetate	0.98
DME	1,2-Dimethoxyethane	0.15
1,3-DX	1,3-Dioxane	0.55
1,4-DX	1,4-Dioxane	0.51
THF	Tetrahydrofuran	0.29
Me-THF	2-Methyltetrahydrofuran	0.23
ETH	Ethoxyethane	0.17
TBME	tert-Butyl methyl ether	0.20
TMOS	Trimethyl methoxysilane	0.33
DMMS	Dimethyldimethoxysilane	0.32
DMOTFS	dimethoxy(methyl)(3,3,3-trifluoropropyl)silane	0.36

Comment 10: The authors attempted to propose general guidance based on the selected system; however, this is not acceptable. More systems should be tested to draw a 'general' conclusion.

Reply: We really appreciate the reviewer’s constructive comment. We recognize the importance of validating our findings across a wider range of systems to establish their broader applicability. The reason why there are fewer types of solvents studied is the compatibility of the solvent with lithium metal, that is, the reduction potential of the solvent. Solvents with high reduction potentials have high reactivity with lithium metal, and it is difficult to achieve high Coulombic efficiency even under high concentration conditions. Therefore, there are more ether solvents in the solvents studied here. By including common solvent families (carbonate, ether, acetate, sulfone, silane), we aim to further test and refine the proposed principles. Notably, among 30 solvents studied in this work (Supplementary Fig. 1), 1h-perfluorohexane (CFH), Methyl perfluorobutyl ether (MNE), and 1,1,1,2,2-Pentafluoro-3-methoxypropane (PMP) also meet the proposed dispersion-dominated design principles, which is compatible with Li metal with long-term cycling CE > 99.6% (Fig. R2, Supplementary Table 5). All of these Li metal efficiencies are higher than the TMS, DME, DMC, and TMP-based localized-high-concentration electrolytes, and much higher than traditional carbonate electrolytes. Such high Li metal CEs verified the universality of our design principles.

Fig. R2 CE statistics for different types of electrolytes, each dot represents an electrolyte.

Supplementary Fig. 1 Molecular structures of solvents studied in this work.

Supplementary Table 5 Li metal CE in Li||Cu cells using different combinations of solvent pairs.

Type	Electrolytes	Mole ratio (LiFSI:1.0)	CE (%) ^a	Test method
------	--------------	------------------------	---------------------	-------------

Type I + Type II	TMP-TTE	1.0:1.25:1.5	98.4	Aurbach
	TMP-FB ₁₂₃	1.0:1.25:1.5	99	Aurbach
	DMC-FB ₁₂₃	1.0:1.25:1.5	99.1	Aurbach
	DME-FB ₁₂₃	1.0:1.25:1.5	99.2	Aurbach
	DMC-TTE	1.0:1.25:1.5	99.3	Aurbach
	DME-TTE	1.0:1.25:1.5	99.3	Aurbach
	TMS-BTFE	1.0:1.25:1.5	98.4	Aurbach
	TMS-FB ₁₂₃	1.0:1.25:1.5	98.8	Aurbach
Type I + Type IV	TMOS-FB ₁₂₄₅	1.0:2.5:3.0	99.3	Aurbach
	TMOS-FB ₁₂₄	1.0:2.5:3.0	99.3	Aurbach
	MeTHF-FB ₁₂₃	1.0:2.5:3.0	99.1	Aurbach
	TBME-FB ₁₂₃	1.0:2.5:3.0	99.2	Aurbach
	TMOS-TTE	1.0:2.5:3.0	99.3	Aurbach
	TBME-BTFE	1.0:2.5:3.0	99.3	Aurbach
Type III + Type IV	MeTHF-FB ₁₃₅	1.0:2.5:3.0	99.65	Aurbach
	TBME-FB ₁₃₅	1.0:2.5:3.0	99.6	Aurbach
	TMOS-FB ₁₃₅	1.0:2.5:3.0	99.7	Aurbach
	TMOS-CFH	1.0:2.5:3.0	99.55	Aurbach
	TBME-MNE	1.0:2.5:3.0	99.57	Aurbach
	MeTHF-CFH	1.0:2.5:3.0	99.6	Aurbach

^a Three cells were measured for calculating CE of each electrolyte.

Some other questions

- 1) The manuscript mentions that the electrolyte LiFSI-TMOS-FB₁₃₅, surpasses commercial electrolytes. Please clarify how this comparison was done? Commercial electrolytes are usually bad with Li metal anodes, so is this a valid comparison? Maybe take as a reference system another electrolyte system compatible with Li metal anodes.

Reply: We really appreciate the reviewer’s constructive comment, which helps us to improve the quality of our paper. We acknowledge that standard commercial electrolytes, such as those based on LiPF₆ in carbonate solvents, are not optimized for lithium metal anodes. Such commercial carbonate electrolyte serves primarily to illustrate the benefits of our approach rather than being a definitive benchmark. To strengthen the validity of our analysis, we have performed additional experiments using a lithium high-concentration electrolyte (LHCE) system based on **1LiFSI-1.2DME-3TTE**, a composition well-documented in the literature for its high performance in Li metal cycling. The cycling stability (Fig. 4c) and pouch cell performances (Fig. 4d) still have advantages over the well-known LiFSI-1.2DME-3.0TTE LHCE system.

Fig. 4 Electrochemical performance of the designed electrolytes. a, Li metal CE in Li||Cu cells for different electrolytes (0.5 mA cm⁻²). **b,** Long-term cycling stability of Li||Cu cells for different electrolytes (0.5 mA cm⁻², 3 mAh cm⁻²). **c,** Long-term cycling stability of 20 μm-Li||4.5 V-LCO

cells at 0.2C/0.5C charge/discharge (LCO loading: 2.3 mAh cm⁻², 1C = 180 mA g⁻¹). **d**, Long-term cycling stability of Cu||NCM523 pouch cells at 0.5C/1C charge/discharge (NCM523 loading: 2.86 mAh cm⁻¹, 1C = 200 mA g⁻¹).

In the revised manuscript, we added the following sentences (Page 16 and 17):

Such cycling stability still has advantages over the well-known LiFSI-1.2DME-3.0TTE LHCE system.

Compared to other Li metal-friendly electrolytes, the anode-free pouch cell using the LiFSI-2.5TMOS-3.0FB₁₃₅ electrolyte achieves 76% CR over 100 cycles owing to favorable Li utilization under fluoride-rich SEI/CEI protection.

2) For the calculation of the Li-solvent and FSI-solvent binding energies, were the initial structures taken from previous simulations or they were built manually? If the structures were taken from previous bulk simulations the results should be more reliable.

Reply: We really appreciate the reviewer's constructive comment. For the calculations of Li⁺-solvent and FSI-solvent binding energies, the initial structures were extracted from previous bulk simulations. The relevant calculation details are supplemented in the *Method* section.

For simulations of Li⁺-solvent and FSI-solvent dimer complexes in bulk solvent, various solvent molecules with LiFSI complexes were used in a periodic cubic box (column is calculated on the basis of density). A 200-ps MD simulation was performed to sample the stable configuration in bulk solvent. The self-consistent charge density-functional tight-binding (SCC-DFTB) implementation was employed using the GFN2-xTB (GBSA) approach in xtb package (<https://github.com/grimme-lab/xtb/>). Geometries of all dimer complexes were extracted from bulk MD simulations, and further optimized using the B3LYP exchange-correlation functional with Grimme's DFT-D3(BJ) empirical dispersion correction.

3) The proposed strategy is a bit similar from the work of 10.1038/s41586-022-05627-8, where a soft solvent is chosen based on a donor number and dielectric constant. Please cite and discuss this work. It is interesting since in that work the “best” solvents had low donor numbers and high dielectric constants.

Reply: We thank the reviewer for pointing out the relevant work (DOI: 10.1038/s41586-022-05627-8). This study indeed provides valuable insights into the role of solvent properties, such as donor number (DN) and dielectric constant (ϵ), in the design of efficient electrolytes for lithium ion batteries. We agree that their finding-that solvents with low DN and high ϵ are particularly effective-is an interesting and important consideration that complements the strategy employed in our work.

In the referenced study, the emphasis is on the selection of "soft solvents" with specific DN and ϵ values to balance ionic dissociation and solvent stability, achieving superior electrochemical performance of Li ion batteries. In our work, we explore the compatibility of the mixed-solvent system (LiFSI-TMOS-FB₁₃₅), focusing on how its molecular interactions and anion aggregation behavior influence SEI formation and LMA stability. The concept of using solvents with low DN and high ϵ aligns with some of the characteristics of TMOS in our electrolyte system. TMOS, with its relatively low DN, likely interacts weakly with Li⁺, which may promote the formation of anion-rich solvation structures, as seen in our simulations. Furthermore, the role of FB₁₃₅ in influencing ionic interactions by the presence of a mediator-like structure complements our observations on anion aggregation and its effect on SEI formation.

We have cited this article and added discussion in Page 10

This can also be verified by the smaller donor number value of TMOS, while FB₁₃₅ with a relatively high dielectric constant is more like a medium that can adjust the solvation structure dynamics.

4) How the concentration Salt: Solvent: Diluent ratio was decided?

Reply: We appreciate the reviewer's interest in understanding the decision-making process behind the Salt: Solvent: Diluent ratio used in our study. The ratio was determined based on experimental optimization. The chosen ratio (Salt: Solvent: Diluent = 1:2.5:3.0) provided the best results in terms of ionic conductivity and efficiency (Fig. SX).

Supplementary Fig. 23 Ionic conductivity and CE of electrolytes with different FB:TMOS ratios.

In the revised manuscript, we added the following sentences:

The ratio of 1:2.5:3.0 was chosen to maximize the ionic conductivity (**Supplementary Fig. 23**).

5) In Fig. 2 it is shown that the LiFSI-TMOS-FB₁₃₅ system forms larger anion aggregates, how this is related to the good compatibility of this electrolyte in LMA.

Reply: The formation of larger anion aggregates is closely related to the electrolyte's interfacial behavior and the properties of the solid electrolyte interphase (SEI). Specifically: The FSI⁻ anion is known to decompose preferentially at the LMA surface, producing inorganic species such as LiF and Li₂S that are essential components of a stable SEI. Larger anion aggregates ensure a localized, high concentration of FSI⁻ near the electrode surface, promoting efficient SEI formation. The aggregation of anions

around Li^+ reduces the activity of TMOS and other solvent molecules, minimizing their direct reduction on the LMA surface. This selectivity helps in forming a more inorganic-rich SEI, which is highly beneficial for compatibility with LMAs. Experimental studies have consistently shown that electrolytes with high concentrations of fluorinated anions, such as FSI^- , form robust SEIs that suppress dendrite growth and improve coulombic efficiency (*Nat Energy*, 2024, 9, 987-998, *Nat Energy*, 2024, 9, 1285-1296). Our findings suggest that the larger anion aggregates in LiFSI-TMOS-FB₁₃₅ contribute to these observed improvements by influencing the initial SEI formation dynamics.

In the revised manuscript, we added the following sentences (Page 10):

Larger and more persistent AGGs ensure a high local concentration of FSI^- , which promotes efficient anion reduction and minimizes solvent reduction, contributing to the formation of a more inorganic-rich SEI, which is highly beneficial for compatibility with LMA.

- 6) Please label the color bars in Fig. 2.

Reply: Thank you for the constructive feedback, which has contributed to enhancing the quality of our manuscript. We have diligently reviewed the paper and added units to the color scale bars in the figures. In the revised manuscript, we have made the required corrections and highlighted them in yellow for clarity.

Fig. 2 Solvation structure and dynamics of the electrolytes. **a**, The average size of solvation clusters in electrolytes, the error bar of size is defined as the difference between the maximum and minimum in the time-averaged values in MD simulation. **b**, Li⁺-FSI⁻ pair existence autocorrelation function of different electrolytes with a cut-off distance of 2.5 Å. **c**, Raman spectra and peak deconvolution results. **d**, Heatmap showing the proportion of corresponding solvation structure. **e**, The intermolecular interaction matrix for LiFSI-2.5DME-3.0FB₁₃₅ (upper left) and LiFSI-2.5TMOS-3.0FB₁₃₅ (bottom right). The normalized probability density (P) of intermolecular connections varies from ~ 0 to 100 as coloring from purple to red. **f**, radial distribution function for N (FSI⁻) versus H (FB₁₃₅) or H (FB₁₂₃) in various electrolytes. **g**, ¹H NMR spectra of LiFSI-2.5TMOS-3.0FB₁₃₅ electrolyte and 2.5TMOS-3.0FB₁₃₅ solvent.

7) Fig. 2 is not well cross-referenced, please review.

Reply: We thank the reviewer for pointing out the issue with the cross-referencing of Figure 2. We carefully reviewed the manuscript and made the necessary corrections to the arrangement of the subfigures to ensure that Figure 2 is appropriately referenced in all relevant parts of the manuscript.

8) There is a writing error “high proportion of AGG proportion”

Reply: We thank the reviewer for pointing out the error. “high proportion of AGG proportion” has been revised to “The highest proportion and longest lifetime of AGGs” at Page 10.

9) In Fig. 3 AIMD shows that the FSI contained in the electrolyte (LiFSI-TMOS-FB₁₃₅) in bulk phase decomposes. Does this mean that the electrolyte is actually not stable in solution and other products are forming?

Reply: Thanks for your constructive comment. The decomposition phenomenon at the bulk phase you mentioned might be related to the small box size (13.7 Å × 13.7 Å × 32.4 Å) used in AIMD simulations. Indeed, in AIMD simulations, the limited simulation box size may lead to local concentration effects, which exacerbate the decomposition behavior of electrolyte molecules (*J. Phys. Chem. C*, 2015, 119, 26828-26839; *J. Phys. Chem. C*, 2017, 121, 182-194). Therefore, this observation does not necessarily imply that the bulk electrolyte is fundamentally unstable under all conditions. To further dispel concerns about the size effect, we performed additional AIMD calculations for different electrolyte systems using large simulation boxes. The Li metal anode was modeled using a 16-layer slab of the (100) Li crystallographic plane and the electrolyte box is expanded to 21.67 Å × 18.39 Å × 64.00 Å. When the electrolyte box is enlarged, no decomposition of FSI and TMOS contained in the

electrolyte is observed in the bulk phase (Fig. RX to RX). Then, the reduction decomposition process and products were observed to elucidate the reduction stability of various molecules in the electrolyte, as well as the origin of SEI between the electrolyte and Li metal anode. Our focus remains on investigating the reactivity of different molecules upon contact with Li metal.

Supplementary Fig. 19 AIMD simulations for LiFSI-2.5DME-3.0FB₁₂₃ electrolyte on Li metal. **a**, Radical distribution functions calculated from initial structure before reaction with Li metal, **b**, Evolution of reaction products during AIMD simulation, **c**, Snapshot revealing the initial structure before reaction with Li metal, **d**, Snapshot revealing the chemical reactions at the Li metal anode-electrolyte interface.

Supplementary Fig. 20 AIMD simulations for $\text{LiFSI-2.5TMOS-3.0FB}_{123}$ electrolyte on **Li metal**. **a**, Radical distribution functions calculated from initial structure before reaction with Li metal, **b**, Evolution of reaction products during AIMD simulation, **c**, Snapshot revealing the initial structure before reaction with Li metal, **d**, Snapshot revealing the chemical reactions at the Li metal anode-electrolyte interface.

Supplementary Fig. 21 AIMD simulations for $\text{LiFSI-2.5TMOS-3.0FB}_{135}$ electrolyte on **Li metal**. **a**, Radical distribution functions calculated from initial structure before reaction with Li metal, **b**, Evolution of reaction products during AIMD simulation, **c**, Snapshot revealing the initial

structure before reaction with Li metal, **d**, Snapshot revealing the chemical reactions at the Li metal anode-electrolyte interface.

10) It is proposed based on the interface model that TMOS decomposition is hindered on the Li surface due to the formation of a rich inorganic SEI layer. However, it is known that observing decomposition of organic molecules on Li surface using AIMD is difficult due to the limitation in the time length of these simulations, around 20 ps. If TMOS decomposition is actually hindered by FSI, then TMOS should decompose when simulated on Li surface without the other components.

Reply: We thank the reviewer for pointing out the issue. We agree that the short simulation timescales (approximately 20 ps) inherent to AIMD pose a challenge for capturing slow decomposition processes, especially for organic molecules like TMOS. In our work, we hypothesize that TMOS decomposition is hindered due to the formation of a robust inorganic SEI layer, which is rich in products from FSI reduction. While AIMD cannot directly observe this long-term inhibition, the simulations provide critical insights into the early-stage interactions between TMOS and the modified lithium surface, which are consistent with this hypothesis. For example, the surface chemistry analysis indicates that the Li-F and Li-S species formed from FSI reduction are likely to create a passivating effect, thereby preventing TMOS decomposition. The reviewer's suggestion to simulate TMOS decomposition on a lithium surface without the presence of FSI⁻ anion-derived components is well-taken (Supplementary Fig. 22a). The simulation results indicated that TMOS showed a higher propensity for decomposition on a pristine lithium surface compared to one with an FSI-modified SEI. This finding aligns with the hypothesis that the FSI-derived SEI alters the surface reactivity, thereby inhibiting TMOS decomposition.

Supplementary Fig. 22 Decomposition simulation and energetics for TMOS and FSI anion. a, TMOS decomposition simulation on Li surface. **b,** energetics for multistep decomposition pathways of TMOS and FSI anion.

Furthermore, we supplemented our findings with thermodynamic calculations to assess the likelihood of Both FSI anion and TMOS decomposition pathways. These methods, while not directly capturing dynamic events, provide complementary evidence supporting our hypothesis. As shown in Supplementary Fig. 22b, the FSI anions form anion-derived SEI by gradually accepting electrons with reaction energy of -3.56 eV and -5.95 eV, the TMOS solvents can also be reduced with the reaction energy of -1.42 eV. We will include additional data and analyses from these control simulations in the revised manuscript to substantiate this claim.

11) Information is missing for MD simulation control parameters such as cutoffs, methods for treating Coulombic forces, timesteps, thermostat and barostat methods etc.

Reply: We thank the reviewer for pointing out the issue of missing details for MD simulation. Providing these control parameters is essential for transparency and reproducibility. Below, we outline the specific parameters used in our simulations: All simulations were run with a time step of 1.0 fs. Temperature and pressure were maintained using a Nosé-Hoover thermostat and barostat as defined in LAMMPS. The particle mesh Ewald method was used to calculate electrostatic interactions, with a

real space cut-off of 1.2 nm and a Fourier spacing of 0.12 nm. A cut-off of 1.2 nm was used for non-bonded Lennard-Jones interactions.

We revised the Supplementary Information to include these details, ensuring that readers have a clear understanding of the methods employed and the reasoning behind the observed temperature and energy trends.

All simulations were run with a time step of 1.0 fs. Temperature and pressure were maintained using a Nosé-Hoover thermostat and barostat as defined in LAMMPS. The particle mesh Ewald method was used to calculate electrostatic interactions, with a real space cut-off of 1.2 nm and a Fourier spacing of 0.12 nm. A cut-off of 1.2 nm was used for non-bonded Lennard-Jones interactions.

Closing remarks

We would like to extend our sincere appreciation to Reviewer #3 for their invaluable contribution of thought-provoking queries and insightful suggestions, which have significantly enhanced the academic significance of our manuscript. It is our utmost hope that the revised version of our manuscript satisfactorily addresses the concerns raised by Reviewer #3.

Detailed responses to the reviewers' comments

Manuscript ID: **NCOMMS-24-53077-T**

Title: A path towards high lithium efficiency for high-energy batteries based on electrolyte interaction motif descriptor

Authors: Ruhong Li, Xiaoteng Huang, Haikuo Zhang, Jinze Wang, Yingzhu Fan, Yiqiang Huang, Jia Liu, Ming Yang, Yuan Yu, Xuezhong Xiao, Yuanzhong Tan, Hao Bin Wu, Liwu Fan, Tao Deng, Lixin Chen, Yanbin Shen, and Xiulin Fan

Reviewer #1 (Remarks to the Author):

The authors have well addressed the reviewer's comments and the manuscript can be accepted now.

Reply: We sincerely appreciate the reviewer's positive feedback and are grateful for your time and effort in evaluating our work.

Reviewer #2 (Remarks to the Author):

After significant revision, it appears that the authors have made significant efforts to improve the manuscript in a number of ways. First, I am grateful for the authors for providing an uncertainty adjusted CE value in their manuscript. This is encouraging and sets a good example for the field. Secondly, the authors have substantially expanded their dataset, which lends more credibility to the enhanced performance provided by their designed system. Lastly, the expanded discussion of how dispersion/coulomb interaction distribution is more precise than solvation energy is improved, though I would suggest a few minor additions (below). In general, I believe these remaining comments to be minor and that the manuscript is in publishable form.

Reply: We sincerely appreciate the reviewer's thoughtful feedback and their recognition of our efforts to improve the manuscript. We are glad to hear that the uncertainty-adjusted CE value, expanded dataset, and refined discussion were well received. Below, we address your suggestions point by point to further refine the manuscript. We thank the reviewer for their time and valuable insights.

Comment 1: The description of these distinctions in the rebuttal are much more detailed in the rebuttal comments than in the manuscript. I recommend that the newly added passages be edited to provide more precise scientific language.

Reply: We have carefully revised the relevant sections in the manuscript to align with the clarity and detail presented in our rebuttal. Specifically, we have refined the wording to ensure more precise scientific language while maintaining readability. We believe these changes enhance the overall clarity and rigor of our discussion.

In the revised *Manuscript*, we modified the following sentences:

Page 4:

In this work, we implement two structurally distinct descriptors for Li^+ and FSI^- species to address their fundamentally divergent solvation behaviors and chemical interactions within the electrolyte system. Li^+ , as a hard Lewis acid, primarily interacts with solvent

molecules through strong electrostatic interactions and directional coordination bonds, necessitating a descriptor that captures its solvation environment and binding strength. The E_{ele}/E_{ind} (Li^+) ratio reflects the relative dominance of electrostatic versus inductive forces in shaping the cation solvation structure. In contrast, anions exhibit more complex solvation dynamics governed by synergistic effects including polarizability, charge delocalization, and ion-pairing tendencies, requiring a distinct descriptor to accurately capture these effects. The E_{ele}/E_{dis} (FSI^-) ratio evaluates the interplay between long-range electrostatic forces and short-range dispersion interactions in anion-solvent interactions, which directly correlates with the anion's ion-cluster dynamics, thereby modulating its reduction susceptibility and subsequent participation in SEI formation processes. The dispersion-dominated FSI^- -solvent interactions stabilize the solvation shell, reduce the dissociation of Li^+ solvation structures, and extend the lifetime of ion pairs, further enhancing the stability and performance of the electrolyte.

Page 6:

The rational design of prospective electrolyte systems necessitates primary selection of electrochemically stable solvents with cathodic compatibility, as quantitatively assessed through their reduction potentials ($E_{\text{reduction}}$ vs. Li^+/Li , **Supplementary Fig. 1 and Supplementary Table 1**).

Page 9:

The efficacy of interaction ratios complemented by energetics analyses allows for a more accurate prediction of ion-cluster dynamics and interfacial regulation mechanisms, offering a promising perspective for electrolyte design.

Page 10:

The formation of stabilized anion aggregation clusters (AGGs) with extended lifetimes (13.6 ps) maintains elevated local FSI^- concentrations, preferentially directing anion-derived SEI formation through selective decomposition pathways. This mechanism effectively suppresses solvent reduction while establishing inorganic-dominated interphases, thereby fundamentally enhancing LMA compatibility.

Page 11:

The radial distribution function (RDF) results demonstrate that the solvation sheath of Li^+ predominantly comprises coordinating FSI^- anions and DME/TMOS solvents (Supplementary Fig. 12). Notably, the pronounced interaction between FB_{135} and anions can be observed in Fig. 2f.

Comment 2: I suggest that the authors provide statistical regressions of CE vs. E_{cell} (solvation energy) and CE vs. $E_{\text{dis}}/(E_{\text{dis}}+E_{\text{ele}})$ (Figure R3) to support their claim that this metric better predicts CE in the weaker solvation regime. The two correlation plots in the current main text and SI are relatively difficult to judge and compare given how many electrolyte titles are packed into the white space.

Reply: We appreciate the reviewer's suggestion to enhance the quantitative support for our claim regarding the predictive power of $E_{\text{dis}}/(E_{\text{dis}}+E_{\text{ele}})$ in the weaker solvation regime. In response, we have now included statistical regression analyses for CE vs. E_{cell} (solvation energy) and CE vs. $E_{\text{dis}}/(E_{\text{dis}}+E_{\text{ele}})$ in Fig. R1. The correlation between CE and $E_{\text{dis}}/(E_{\text{dis}}+E_{\text{ele}})$ ratio is relatively stronger (coefficient of determination (R-square) = 0.855, Pearson correlation coefficient (Pearson's r) = 0.925), especially for most weakly solvated electrolyte systems ($E_{\text{cell}} < -60$ mV). These analyses provide a clearer comparison of the predictive strength of these metrics.

Fig. R1 Relationship of electrolytes composed of different types of solvents. a, E_{cell} and CE, b, dispersion contribution ratio and CE.

Additionally, we acknowledge the reviewer's concern regarding the readability of the correlation plots. To address this, we have adjusted the formatting to reduce visual clutter, ensuring that individual data points are more distinguishable. Specifically, we increased spacing and used vertical marker styles. We believe these modifications improve clarity while maintaining the integrity of the data presentation.

In the revised *Manuscript*, we modified the following figure:

Fig. 1 Electrolyte design strategies. **a**, Schematic illustration of the ion-solvent interaction regulated solvation structure. **b**, The molecular design diagram of E_{ele}/E_{dis} (FSI) versus E_{int} (Li^+). **c**, The molecular design diagram of E_{ele}/E_{dis} (FSI) versus E_{ele}/E_{ind} (Li^+). Molecular structures and corresponding abbreviations of the investigated solvents are shown in Supplementary Fig. 1. **d**, Relationship between the dispersion contribution ratio and Coulombic efficiency of electrolytes composed of different type of solvents, schematic interactions of the diluent-FSI-solvent pairs (left is FB_{123} -FSI-DME, right is FB_{135} -FSI-TMOS, respectively) and corresponding ratios of three non-covalent interaction (NCI) components, which was obtained by processing data in Supplementary Fig. 4.

In the revised *Supplementary Information*, we modified the following figure:

Supplementary Fig. 5 Solvation energy and dispersion contribution ratios of different high-performance electrolytes. a, Solvation energies of various electrolytes, b, Relationship of E_{cell} and CE, c, Relationship of dispersion contribution ratio and CE. The more negative the battery potential, the weaker the Li^+ solvation effect.

Comment 3: It would be useful to discuss potential follow-up work regarding how this metric correlates to other electrolyte features such as ionic conductivity, transference number, ionicity, etc. though I agree it is beyond the scope of this work.

Reply: We appreciate the reviewer's insightful suggestion regarding potential follow-up work. We agree that this would be a valuable avenue for future research. By refining the solvent-ion interaction parameters, we can more effectively capture the dynamic coordination state of anions and cations, as well as their molecular-scale correlations. This deeper understanding is crucial for elucidating key ion transport properties, including conductivity and transference number. To acknowledge this point, we have added a brief discussion in the manuscript, outlining how these correlations could further enhance the understanding and predictive power of our approach. We thank the reviewer for highlighting this aspect, as it helps position our work within a broader research context.

In the revised manuscript, we added the following sentences (Page 24):

Moreover, the quantitative determination of solvent-ion interaction parameters enables precise characterization of the dynamic coordination equilibrium between anions and cations, along with their time-dependent characteristics. This mechanistic insight provides a critical fundamental understanding of the structure-property relationships governing ionic transport phenomena, particularly in elucidating the quantitative dependence of macroscopic electrochemical parameters (such as ionic conductivity and Li^+ transference number) on the molecular-scale solvation dynamics.

Reviewer #3 (Remarks to the Author):

The authors have addressed my questions and concerns thoroughly in the response letter. However, I believe these important details should also be better elaborated upon in the manuscript—particularly in the introduction—to provide the audience with stronger background knowledge regarding the focus on the two selected descriptors.

Currently, the introduction has not been significantly updated, and the motivation for the study remains somewhat vague, even though the response letter provides better explanations. For instance, why were the two different descriptors chosen for Li and the anion, and what is the underlying physical rationale? Additionally, why is understanding anion-solvent interactions critical in this system? These points are fundamental to the study and should be explained in greater detail to enhance the manuscript's clarity and impact. Besides this, I think the research is acceptable.

Reply: We sincerely appreciate the reviewer's positive assessment of our revisions. We acknowledge the need to further elaborate on the motivation behind our choice of descriptors and the significance of anion-solvent interactions in the manuscript itself. These revisions aim to enhance the manuscript's clarity and ensure that readers fully understand the study's motivation and context.

(1) First, we employ two distinct descriptors for Li⁺ and FSI⁻ anion to account for their fundamentally different solvation behaviors and chemical interactions within the electrolyte system. Li⁺, as a hard Lewis acid, primarily interacts with solvent molecules through strong electrostatic interactions and directional coordination bonds, necessitating a descriptor that captures its solvation environment and binding strength. The E_{ele}/E_{ind} (Li⁺) ratio captures the relative contributions of electrostatic and induction forces governing the local solvation environment. A higher E_{ele}/E_{ind} (Li⁺) ratio may correlate with weaker coordination of Li⁺ to anions. In contrast, anions exhibit more complex solvation dynamics governed by synergistic effects including polarizability, charge delocalization, and ion-pairing tendencies, requiring a distinct descriptor to accurately capture these effects. The E_{ele}/E_{dis} (FSI⁻) ratio evaluates the interplay between long-range electrostatic forces and short-range dispersion interactions in anion-solvent interactions, which directly correlates with the anion's ion-

cluster dynamics, thereby modulating its reduction susceptibility and subsequent participation in SEI formation processes. As demonstrated in our work, A higher E_{ele}/E_{dis} (FSI) ratio can prolong the lifetime of Li-FSI ion pairs, leading to greater anion-derived SEI contributions and improved CE. This comprehensive approach ensures a more holistic understanding of the electrolyte system, which is critical for optimizing performance.

(2) Additionally, we have expanded our discussion on anion-solvent interactions to better highlight their critical influence on system behavior. The Li^+ interactions within the solvation structure are primarily governed by electrostatic forces, with the additional presence of FSI-solvent interactions playing a key role in stabilizing the solvation shell. This stabilization effect reduces the dissociation of the Li^+ solvation structure, which, in turn, prolongs the lifetime of ion pairs (as illustrated in Supplementary Fig. 8). Furthermore, we emphasize the formation of stabilized anion aggregation clusters (AGGs) with extended lifetimes—specifically, 13.6 ps for the TMOS-FB₁₃₅ system. These stable AGGs help maintain elevated local FSI⁻ concentrations, which preferentially direct anion-derived SEI formation through selective decomposition pathways. This mechanism is pivotal in suppressing solvent reduction and promoting the formation of inorganic-dominated interphases, thereby fundamentally enhancing lithium metal anode (LMA) compatibility.

We have substantially revised the *Introduction* (Page 4-5) to provide a clearer rationale for selecting different descriptors for Li and the anion, emphasizing their distinct roles in capturing key physicochemical properties.

In this work, we implement two structurally distinct descriptors for Li^+ and FSI⁻ species to address their fundamentally divergent solvation behaviors and chemical interactions within the electrolyte system. Li^+ , as a hard Lewis acid, primarily interacts with solvent molecules through strong electrostatic interactions and directional coordination bonds, necessitating a descriptor that captures its solvation environment and binding strength. The E_{ele}/E_{ind} (Li^+) ratio reflects the relative dominance of electrostatic versus inductive forces in shaping the cation solvation structure. In contrast, anions exhibit more complex solvation dynamics governed by synergistic effects including polarizability, charge delocalization, and ion-pairing tendencies, requiring a distinct descriptor to

accurately capture these effects. The E_{ele}/E_{dis} (FSI⁻) ratio evaluates the interplay between long-range electrostatic forces and short-range dispersion interactions in anion-solvent interactions, which directly correlates with the anion's ion-cluster dynamics, thereby modulating its reduction susceptibility and subsequent participation in SEI formation processes. The dispersion-dominated FSI⁻-solvent interactions stabilize the solvation shell, reduce the dissociation of Li⁺ solvation structures, and extend the lifetime of ion pairs, further enhancing the stability and performance of the electrolyte.

In the revised *Manuscript*, we modified the following sentence (Page 10):

The formation of stabilized anion aggregation clusters (AGGs) with extended lifetimes (13.6 ps) maintains elevated local FSI⁻ concentrations, preferentially directing anion-derived SEI formation through selective decomposition pathways. This mechanism effectively suppresses solvent reduction while establishing inorganic-dominated interphases, thereby fundamentally enhancing LMA compatibility.